# Enhanced role of the entorhinal cortex in adapting to increased working memory load

Jiayi Yang [1], Dan Cao [1] ✉, Chunyan Guo[1], Lennart Stieglitz [2], Debora Ledergerber[3], Johannes Sarnthein [2,4] & Jin Li [1] ✉

In daily life, we frequently encounter varying demands on working memory (WM), yet how the brain adapts to high WM load remains unclear. To address this question, we recorded intracranial EEG from hippocampus, entorhinal cortex (EC), and lateral temporal cortex (LTC) in humans performing a task with varying WM loads (load 4, 6, and 8). Using multivariate machine learning analysis, we decoded WM load using the power from each region as neural features. The results showed that the EC exhibited both higher decoding accuracy on medium-to-high load and superior cross-regional generalization. Further analysis revealed that removing EC-related information significantly reduced residual decoding accuracy in the hippocampus and LTC. Additionally, we found that WM maintenance was associated with enhanced phase synchronization between the EC and other regions. This inter-regional communication increased as WM load rose. These results suggest that under higher WM load, the brain relies more on the EC, a key connector that links and shares information with the hippocampus and LTC.

Working memory (WM) is a fundamental cognitive function that enables the temporary storage and manipulation of information[1]. It plays a critical role in everyday life, supporting a range of cognitive activities from basic tasks to complex problem-solving[2]. WM demands vary significantly depending on the amount of information to be managed. For instance, maintaining a few appointments requires minimal cognitive effort, whereas managing multiple tasks, times, and details places greater demands on WM. This raises an important question: how does the brain flexibly adapt to different levels of WM load?

Previous research has explored how neural activity changes across different load levels[3–5]. Findings are mixed, with some studies suggesting a linear increase in neural activity with increasing load[6], others indicating an initial increase followed by a plateau[7], and still others proposing an inverted U-shaped response[8]. These findings indicated that the transition from low to medium load shows relatively consistent patterns of increased activity across WM regions. However, the response from medium-to-high load is less consistent, suggesting

the need for more nuanced investigations that separately examine changes from low-to-medium load and from medium-to-high load. Furthermore, previous studies have primarily focused on the relationship between neural activity in individual brain regions and WM load, examining each region separately. Here, we examine how the contributions of different brain regions vary with increasing load and assess the role of inter-regional connectivity.

Recent studies have begun to explore the involvement of non-traditional WM areas, such as the medial temporal lobe (MTL), in WM maintenance[9]. However, the precise role of MTL regions, particularly in relation to WM load, remains unclear. Within the MTL, the hippocampus has received considerable attention. Studies have shown that hippocampal neural activity allows accurate decoding of WM content[10] and stably maintains information in the absence of the stimulus[11]. Interestingly, hippocampal neural activity increases from low-to-medium load but plateaus from medium-to-high load, suggesting the involvement of alternative mechanisms at higher demands[12]. Beyond the hippocampus, other MTL regions, such as the entorhinal cortex

[1]School of Psychology, Capital Normal University, Beijing, China. [2]Department of Neurosurgery, University Hospital Zurich, University of Zurich, Zurich, Switzerland. [3]Swiss Epilepsy Center, Clinic Lengg, Zurich, Switzerland. [4]Zurich Neuroscience Center, ETH Zurich, Zurich, Switzerland. ✉ e-mail: 7163@cnu.edu.cn; 7049@cnu.edu.cn

(EC), have also been reported to play a role in WM processing. EC neurons exhibit persistent firing during WM task delays[12,13], and both firing rates in the EC[13] and inter-regional communication between the EC and the hippocampus[14] tend to increase with WM load. Beyond the MTL, the lateral temporal cortex (LTC) has also been implicated in WM-related gamma activity changes[15] and exhibits bidirectional oscillatory interactions with the hippocampus[16] during WM processing. Despite these findings, direct studies simultaneously investigating activity in the EC, hippocampus, and LTC are lacking, leaving open questions about how these regions coordinate to maintain the increased WM content.

To address these gaps, we simultaneously recorded intracranial EEG (iEEG) from the EC, hippocampus, and LTC in 13 epilepsy patients performing a WM task (Fig. 1A). We investigated neural adaptations to WM load by machine learning decoding analyses under low-to-medium and medium-to-high load conditions separately. Decoding analyses were conducted using single-regional power features (Fig. 1B), cross-regional decoding generalization (Fig. 1C), and residual-based decoding to remove EC-shared activity (Fig. 1D). In addition, functional connectivity analysis (Fig. 1E) was used to assess load-dependent changes in inter-regional connectivity. Our findings indicate that the EC exhibits superior decoding performance on medium-to-high WM load, shares load-related information with other regions, and displays load-dependent increases in its functional connectivity with the hippocampus and LTC.

## Results
### Behavioral Results
Thirteen patients with drug-resistant epilepsy (6 females) participated in the WM task. The average memory accuracy across all participants was $92.04 \pm 3.35\%$ (median 92.5% correct trials), indicating that all patients scored well in the task. Across all participants, the median memory capacity was 6.5 (range: 5.6–7.5) based on Pashler's $K_P$, which indicates that the participants were able to maintain about 6 letters in memory and that load exceeded capacity in high load trials (load 8). In sum, these results show that all participants were able to perform the task.

### Higher decoding accuracy of EC under medium-to-high demands
Next, we examined the roles of different brain regions in adapting to varying loads at a fine-grained scale. Compared to univariate analysis, multivariate decoding approaches can capture information that is lost when averaging signals, providing greater sensitivity for detecting differences between conditions[17]. Therefore, we used multivariate analysis to decode load 4 vs load 6 and load 6 vs load 8 in the subsequent analyses separately. The maintenance of WM information elicited z-scored power changes in the 1–40 Hz frequency range. Accordingly, neural activity within this range was included in the subsequent multivariate decoding analyses.

We first investigated how the neural activity of the hippocampus, EC, and LTC adapts to increasing cognitive demands during WM maintenance. We applied a linear SVM classifier to decode WM load (load 4 vs load 6, load 6 vs load 8), using power features from the hippocampus, EC, and LTC separately (Fig. 2C). To assess the statistical significance of the decoding results, we created a null distribution of the decoding accuracy by shuffling the relationship between labels and data 100 times. Decoding accuracy exceeding the 95% threshold of this null distribution was considered statistically significant. The results showed that the decoding accuracies on low-to-medium and medium-to-high load conditions using power features from the hippocampus, EC, and LTC were significantly above chance level. The detailed 95% threshold values are provided in Supplementary Table S2. Next, to assess differences in decoding accuracy among regions, we performed permutation t-tests. As

shown in Fig. 2D, there were no significant differences in decoding accuracy on low-to-medium load condition among the hippocampus (mean ± S.D.: $56.33 \pm 3.94\%$), EC ($56.44 \pm 1.10\%$), and LTC ($56.08 \pm 3.84\%$; permutation $t$ test: EC vs hippocampus: $t = 0.29$, $p = 0.768$; EC vs LTC: $t = 0.90$, $p = 0.379$; LTC vs hippocampus: $t = -0.42$, $p = 0.690$). However, under medium-to-high load condition, the EC exhibited the highest decoding accuracy ($55.49 \pm 1.72\%$), and the decoding accuracy from the LTC power features ($53.82 \pm 2.10\%$) was significantly higher than the hippocampus power features ($52.84 \pm 1.67\%$; EC vs hippocampus: $t = 11.16$, $p < 0.001$; EC vs LTC: $t = 6.86$, $p < 0.001$; LTC vs hippocampus: $t = 3.96$, $p < 0.001$). We further calculated the difference in decoding accuracy between medium-to-high and low-to-medium load conditions. As shown in Fig. 2E, the EC exhibited the smallest change in decoding accuracy ($0.96 \pm 2.17\%$; hippocampus vs EC: $t = 5.58$, $p < 0.001$; LTC vs EC: $t = 2.82$, $p = 0.005$), while the hippocampus ($3.48 \pm 4.15\%$) showed a greater change than the LTC ($2.26 \pm 4.20\%$; hippocampus vs LTC: $t = 2.02$, $p = 0.047$). These findings suggest that neural activity from the EC exhibits a greater capacity to adapt to increasing load.

As noted in previous studies, the EC bridges the information transfer between the hippocampus and cortex[18]. Therefore, we hypothesize that the EC, sharing information from both the hippocampus and the LTC, is consequently more sensitive to increasing WM load. To test this hypothesis, we employed two approaches: cross-regional decoding and residual-based decoding.

### Higher cross-regional generalization of EC under medium-to-high demands
We first conducted a cross-regional decoding analysis to assess information sharing between brain regions (schematic in Fig. 3A). Our hypothesis was that if two regions share load-sensitive information, a decoding model trained on one region should perform well when tested on another. For example, to test the EC's information sharing with other regions, we trained a classifier using power features from the EC and tested its performance on hippocampal features. We systematically assessed cross-regional generalization by applying EC-trained models to the hippocampus and LTC and, as a control, testing hippocampus- and LTC-trained models on the other regions. This analysis was conducted separately for low-to-medium and medium-to-high load conditions. Permutation t-tests were used to assess whether cross-regional generalization significantly differed among the three brain regions.

The cross-regional decoding accuracies on low-to-medium and medium-to-high load conditions across hippocampus, EC, and LTC were significantly above chance level (see Supplementary Table S2 for details). Under low-to-medium load condition, there was no significant difference in cross-regional decoding accuracy among the hippocampus (Fig. 3C; $57.14 \pm 1.83\%$), EC ($57.25 \pm 1.14\%$), and LTC ($57.51 \pm 1.47\%$; EC vs hippocampus: $t = 0.47$, $p = 0.637$; EC vs LTC: $t = -1.54$, $p = 0.128$; LTC vs hippocampus: $t = 1.44$, $p = 0.141$). However, under medium-to-high load conditions, the EC exhibited significantly higher cross-regional decoding accuracy ($53.13 \pm 2.61\%$) compared to the hippocampus ($50.45 \pm 2.25\%$; EC vs hippocampus: $t = 8.34$, $p < 0.001$) and the LTC ($51.97 \pm 1.97\%$; EC vs LTC: $t = 3.47$, $p = 0.002$). In addition, the LTC showed significantly greater generalization than the hippocampus ($t = 5.47$, $p < 0.001$). These findings demonstrate that under medium-to-high cognitive demands, the EC exhibits the highest cross-regional generalization.

### Significant decoding accuracy reduction after removing EC information
Next, we conducted a residual-based decoding analysis. We hypothesized that if the EC shares WM load-related information with other regions, then removing EC-shared information from the hippocampus and LTC would impact their decoding accuracy. Specifically, we

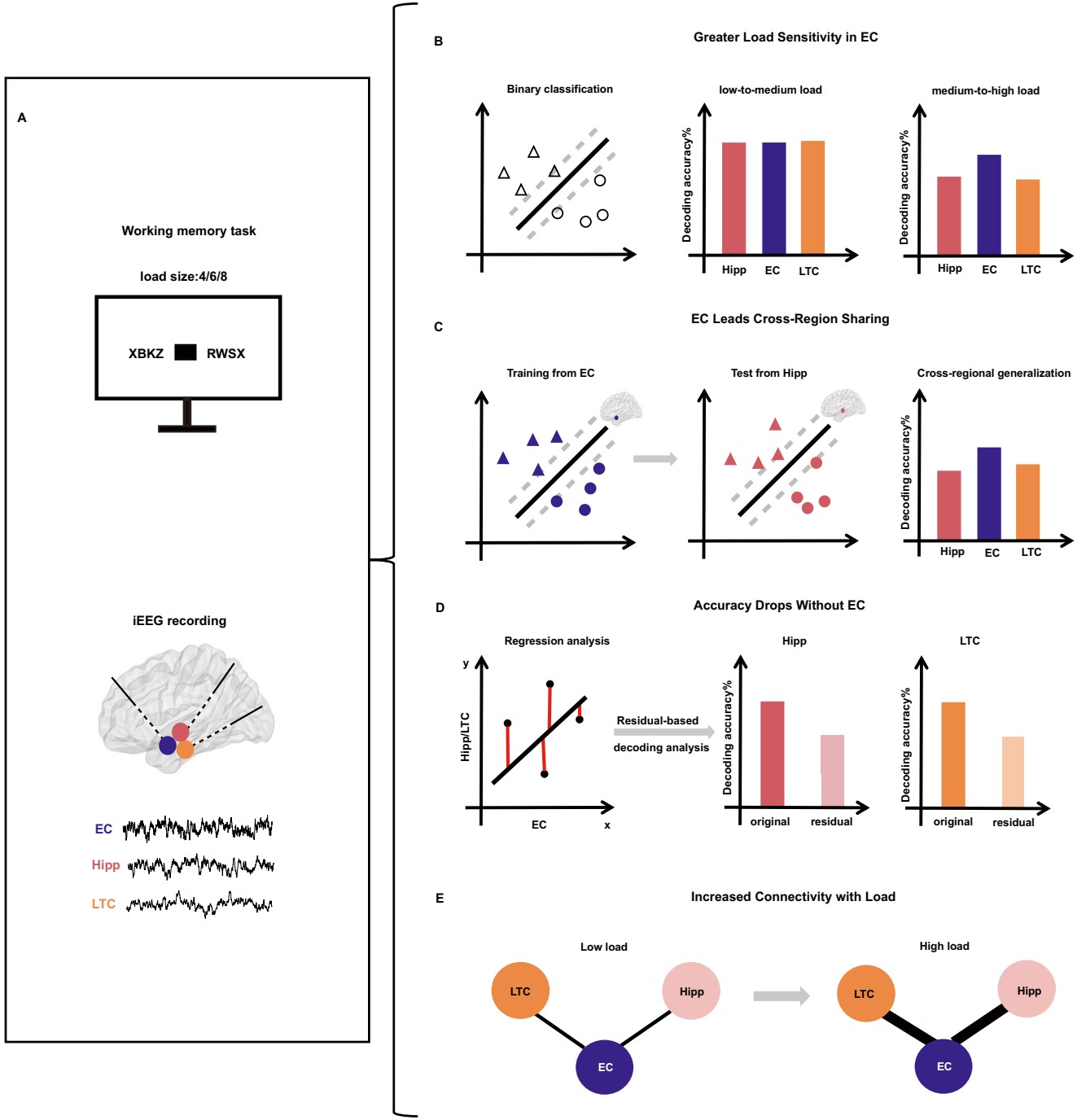

**Fig. 1 | Overview of the analysis pipeline. A** Schematic illustration of the WM task and intracranial EEG (iEEG) recording sites in the entorhinal cortex (EC), hippocampus (Hipp), and lateral temporal cortex (LTC). **B** Under medium-to-high load conditions, decoding accuracy based on EC power features was higher than that derived from the hippocampus or LTC. **C** Cross-regional decoding, in which decoders trained on one region's data were tested on another, revealed that EC-based decoders demonstrated the highest generalization under medium-to-high load conditions. **D** Residual decoding analysis showed that removing neural activity shared with the EC significantly reduced decoding accuracy in the hippocampus and LTC under medium-to-high load conditions. **E** Functional connectivity analysis indicated that the phase locking value (PLV) between the EC and other regions increased with enhanced WM load. The brain (**A**, **C**) was visualized by the BrainNet Viewer toolbox (www.nitrc.org/projects/bnv/)[43].

constructed linear regression models using EC features as independent variables (x) and the hippocampus and LTC features as dependent variables (y), separately for each load. The residuals from these models, representing hippocampus and LTC activity with EC contributions removed, were then used as features to train and test the SVM classifier (Fig. 4A). We expected a reduction in decoding accuracy for the hippocampus and LTC when shared information from the EC was excluded. The permutation $t$ tests were employed to compare

decoding accuracies between the original and residual-based features. As shown in Fig. 4B, the residual-based decoding accuracies were significantly higher than chance level for the hippocampus ($50.34 \pm 1.33\%$) and the LTC ($50.31 \pm 1.41\%$; see Supplementary Table S2 for details). Moreover, the residual-based decoding accuracies were significantly lower than those obtained using the original features from the hippocampus ($t = 11.10$, $p < 0.001$) and LTC ($t = 14.01$, $p < 0.001$), confirming our hypothesis.

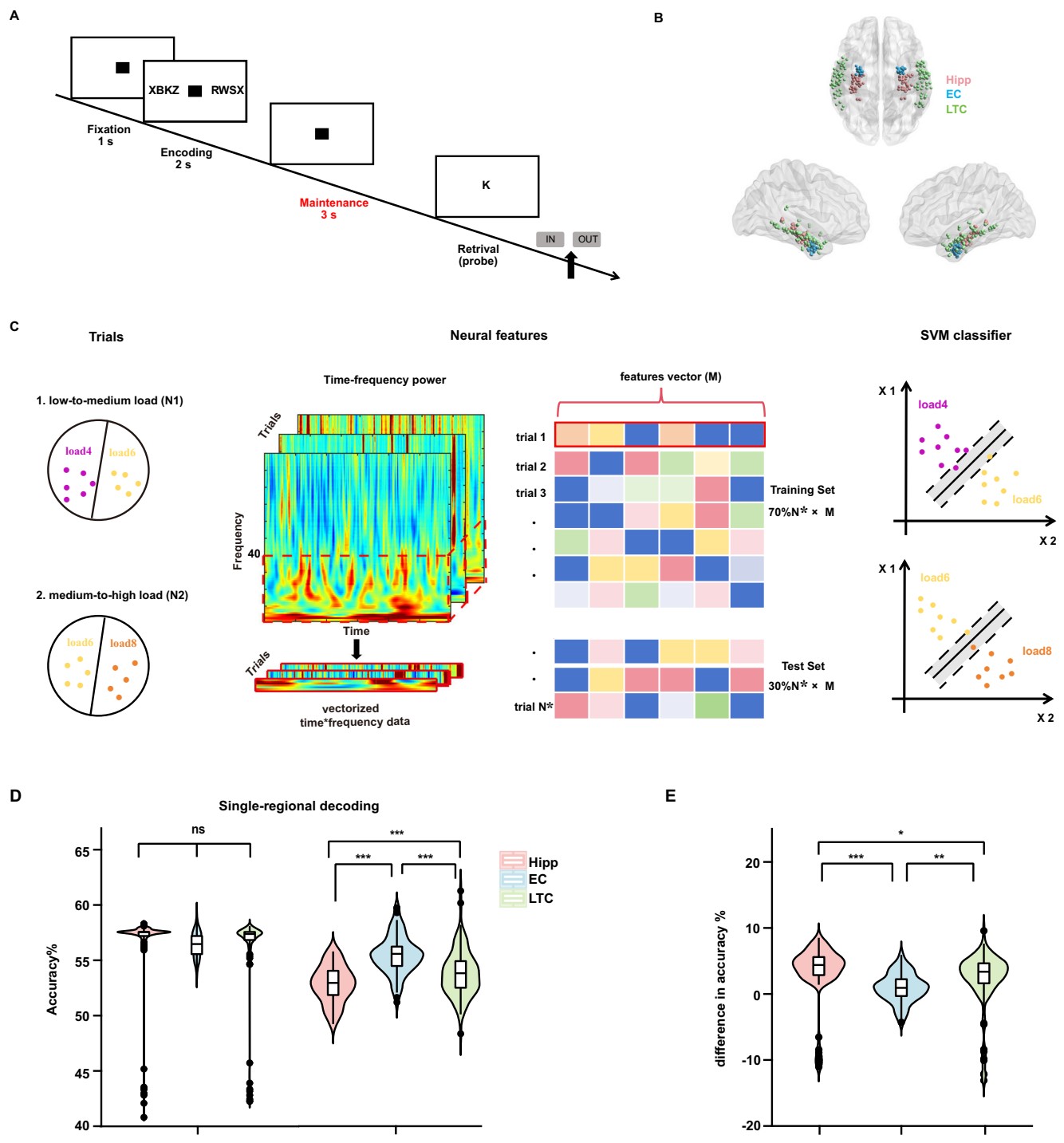

**Fig. 2 | Experimental paradigm, recording sites, schematic, and results of single-regional decoding analysis. A** Each trial began with a 1 s fixation screen, followed by a 2 s presentation of four, six, or eight letters. After letters disappeared, there was a 3 s maintenance period with a black square shown. Participants responded whether a probe letter was part of the original set by pressing "IN" or "OUT". **B** Channel locations of all participants included 91 channels in the hippocampus (Hipp; light red), 46 channels in the entorhinal cortex (EC; light blue), and 136 channels in the lateral temporal cortex (LTC; light green). The brain was visualized by the BrainNet Viewer toolbox (www.nitrc.org/projects/bnv/)[43]. **C** We conducted binary classification (load 4 vs load 6, load 6 vs load 8). Time-frequency analysis was performed on each trial to obtain power spectra in the hippocampus, EC, and LTC. For each classification task, 70% of the data was used for training and 30% for testing with a linear SVM classifier. **D** The decoding accuracy for load 4 vs load 6 did not show significant differences among the hippocampus, EC, and LTC

across all cross-validations ($n = 100$, two-sided permutation $t$ test: EC vs hippocampus: $p = 0.768$; EC vs LTC: $p = 0.379$; LTC vs hippocampus: $p = 0.690$; see distribution with 150 iterations in Supplementary Fig. S1). The EC exhibited the highest decoding accuracy for load 6 vs load 8 ($n = 100$ cross-validations; two-sided permutation $t$ test: all $ps < 0.001$). ***$p < 0.001$. **E** Differences in decoding accuracy between low-to-medium and medium-to-high load conditions were smallest in the EC ($n = 100$ cross-validations; two-sided permutation $t$ test: hippocampus vs EC: $p < 0.001$; LTC vs EC: $p = 0.005$; hippocampus vs LTC: $p = 0.047$). *$p < 0.05$, **$p < 0.01$, ***$p < 0.001$. In the box plots shown in (**D, E**), the center line represents the median, and the edges of the box correspond to the lower and upper quartiles, respectively. The whiskers extend to the minimum and maximum data points at most 1.5 times the interquartile range. Source data are provided as a Source Data file.

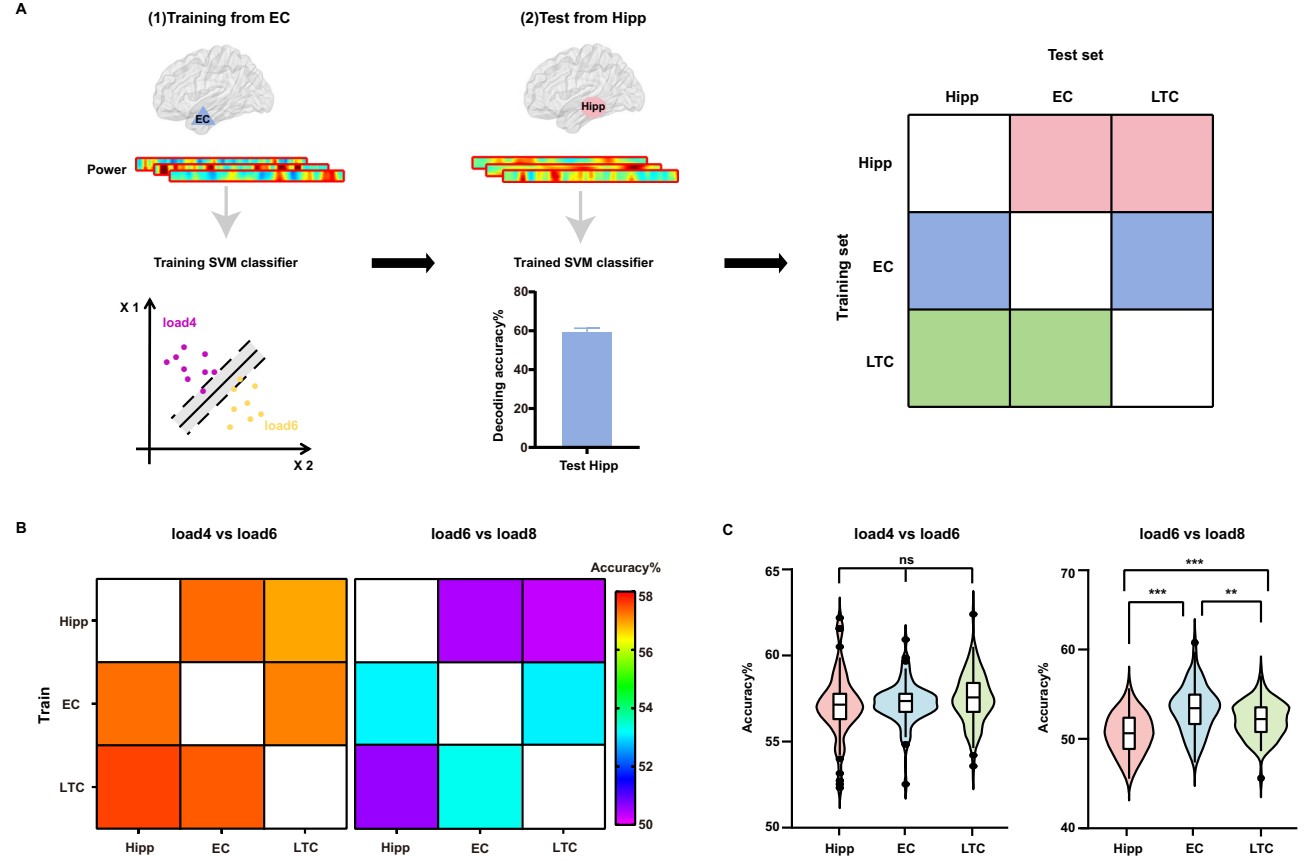

**Fig. 3 | Highest cross-regional generalization in the EC. A** Schematic of cross-regional decoding analysis. Using the entorhinal cortex (EC) as an example, we trained classifiers using power features from EC for each trial and predicted the load using power from the hippocampus (Hipp) for both low-to-medium and medium-to-high load conditions. The specific decoding steps were the same as shown in Fig. 2C. For all brain regions, models were trained using their own power features and tested on data from the other two brain regions. The generalization of each brain region was determined by averaging its accuracy when tested on data from the other two brain regions (hippocampus: light red; EC: light blue; lateral temporal cortex: LTC, light green). The brain was visualized by the BrainNet Viewer toolbox (www.nitrc.org/projects/bnv/)[43]. **B** Accuracy matrix of cross-regional decoding on low-to-medium load (left) and medium-to-high load (right). The rows of the matrix represented the regions used for training, while the columns denoted

the regions employed for testing, with the values representing the average accuracy. **C** The averaged cross-regional decoding accuracy for load 4 vs load 6 did not differ significantly among hippocampus, EC, and LTC across all cross-validations ($n = 100$; two-sided permutation $t$ tests: EC vs hippocampus: $p = 0.637$; EC vs LTC: $p = 0.128$; LTC vs hippocampus: $p = 0.141$). The EC showed the highest cross-regional decoding accuracy for load 6 vs load 8 across all cross-validations ($n = 100$; two-sided permutation $t$ tests: EC vs hippocampus: $p < 0.001$; EC vs LTC: $p = 0.002$; LTC vs hippocampus: $p < 0.001$). **$p < 0.01$, ***$p < 0.001$. The center line represents the median, and the edges of the box correspond to the lower and upper quartiles, respectively. The whiskers extend to the minimum and maximum data points at most 1.5 times the interquartile range. Source data are provided as a Source Data file.

Although we observed a significant drop in decoding accuracy after removing EC-shared information, this effect could potentially be due to the residualization method itself rather than the contribution of the EC. To test whether the observed accuracy decrease was specific to the EC, we conducted a control analysis. Using the same method, we regressed LTC activity from the hippocampus (LTC-residualed) and hippocampus activity from the LTC (Hipp-residualed), then used the resulting residuals for decoding. The analysis showed that decoding accuracies remained significantly above chance level for LTC-residualed ($51.36 \pm 1.22\%$) in the hippocampus and the Hipp-residualed ($50.78 \pm 1.48\%$; Supplementary Table S2) in the LTC. We then compared the reduction in decoding accuracy when EC information was removed ($Hipp_{res-EC}$ - $Hipp_{orig}$) with the reduction value when LTC information was removed ($Hipp_{res-LTC}$ - $Hipp_{orig}$) in the hippocampus. Similarly, we compared the change in decoding accuracy based on EC-residuals ($LTC_{res-EC}$ - $LTC_{orig}$) with that based on Hipp-residuals ($LTC_{res-Hipp}$ - $LTC_{orig}$) in the LTC. This comparison allowed us to assess whether removing EC-shared information had a uniquely strong impact on decoding performance. Results showed

that the $\Delta$ accuracy for EC-residualed was significantly lower than that in LTC-residualed in hippocampus (Fig. 4C; $Hipp_{res-EC}$ - $Hipp_{orig}$: $-2.50 \pm 2.25\%$; $Hipp_{res-LTC}$ - $Hipp_{orig}$: $-1.48 \pm 1.90\%$; $t = -5.32$, $p < 0.001$). Similarly, in the LTC, $\Delta$ accuracy in EC-residualed was also significantly lower than that in Hipp-residualed ($LTC_{res-EC}$ - $LTC_{orig}$: $-3.51 \pm 2.51\%$; $LTC_{res-Hipp}$ - $LTC_{orig}$: $-3.04 \pm 2.74\%$; $t = -2.21$, $p = 0.029$;). These findings indicate that removing EC-shared information has a significantly stronger impact on decoding accuracy in both the hippocampus and LTC.

## EC Activity Pattern Linked to Better Memory Performance

Finally, we examined whether EC properties have functional effects on memory performance. We divided participants into two groups based on their accuracy in recalling items at load 8, categorizing them as either low-performing or high-performing (Fig. 5A). For each group, we conducted single-regional decoding and cross-regional decoding analyses under medium-to-high load conditions. The specific procedures were consistent with those described in the **Methods** section, with participants grouped according to

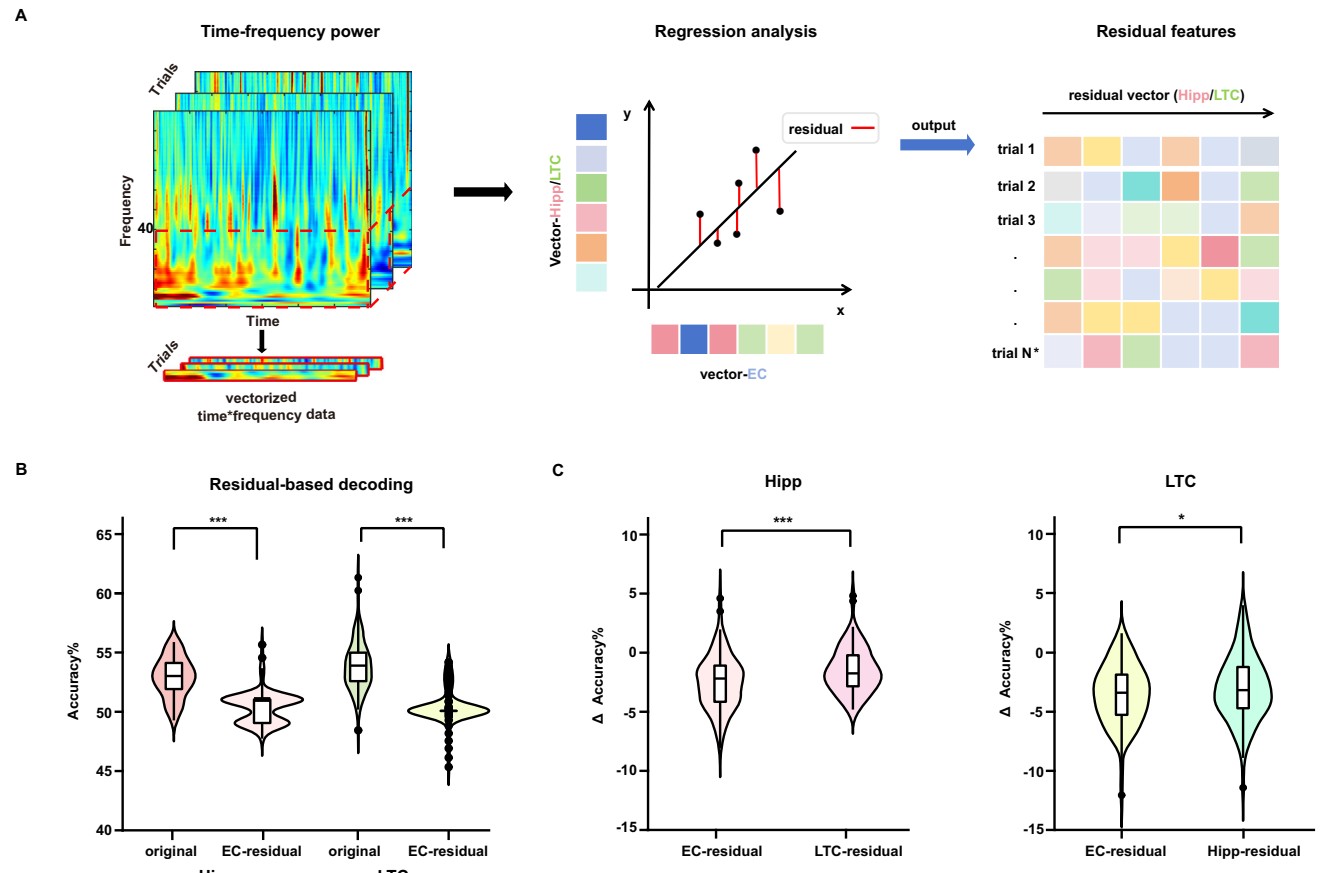

**Fig. 4 | Residual-based decoding analysis. A** Schematic of residual-based decoding analysis. We first transformed the time-frequency power features of each trial across three brain regions into vectors. Then, we used the feature vector of the hippocampus (Hipp) and lateral temporal cortex (LTC) as dependent variables (y) separately, with the features of the entorhinal cortex (EC) as independent variables (x), to construct linear regression models for each trial. The resulting residuals of the hippocampus and LTC were retained as features for training and testing the classifier. The specific decoding steps were the same as shown in Fig. 2C. **B** Original features (dark colors) represented decoding accuracy using the original power features from the hippocampus (red) and LTC (green). EC-residual (light colors) indicated decoding accuracy using residuals after removing EC-shared information from the hippocampus (red) and LTC (green). The EC-residual decoding accuracies were significantly lower than those obtained using the original features from the hippocampus and LTC (n = 100 cross-validations; two-sided permutation $t$ tests: all

ps < 0.001). ***$p$ < 0.001. **C** Left panel: the change in decoding accuracy after removing shared information from EC (EC-residual = Hipp$_{res-EC}$ - Hipp$_{orig}$; light red) and LTC (LTC-residual = Hipp$_{res-LTC}$ - Hipp$_{orig}$; dark red) in the hippocampus. The decrease in decoding accuracy was significantly greater for EC-residual than for LTC-residual (n = 100 cross-validations; two-sided permutation $t$ tests: $p$ < 0.001). Right panel: the change in decoding accuracy after removing shared information from EC (EC-residual = LTC$_{res-EC}$ - LTC$_{orig}$; light green) and hippocampus (Hipp-residual = LTC$_{res-Hipp}$ - LTC$_{orig}$; dark green) in the LTC. The decrease in decoding accuracy was significantly greater for EC-residual than for Hipp-residual (n = 100 cross-validations; two-sided permutation $t$ tests: $p$ = 0.029). *$p$ < 0.05, ***$p$ < 0.001. In the box plots shown in (**B**, **C**), the center line represents the median, and the edges of the box correspond to the lower and upper quartiles, respectively. The whiskers extend to the minimum and maximum data points at most 1.5 times the interquartile range. Source data are provided as a Source Data file.

performance. Permutation tests were used to compare decoding accuracy between the two groups.

The single-regional and cross-regional decoding accuracies were significantly above chance level in both the high- and low-performing groups for the EC (low-group: single-regional 53.21 ± 3.24%, cross-regional 52.57 ± 2.17%; high-group: single-regional 56.22 ± 2.87%, cross-regional 54.39 ± 2.33%), hippocampus (low-group: single-regional 53.89 ± 3.16%, cross-regional 51.53 ± 2.81%; high-group: single-regional 54.12 ± 3.32%, cross-regional 51.95 ± 2.46%), and LTC (low-group: single-regional 52.18 ± 3.47%, cross-regional 51.64 ± 2.72%; high-group: single-regional 51.76 ± 4.60%, cross-regional 51.36 ± 2.13%; Supplementary Table S2). The further comparison revealed that single-regional (Fig. 5B) and cross-regional (Fig. 5C) decoding accuracies were significantly higher for the high-performing group compared to the low-performing group in the EC (single-regional: $t$ = 7.15, $p$ < 0.001; cross-regional: $t$ = 5.94, $p$ < 0.001), whereas decoding accuracies did not differ significantly between performance groups in the hippocampus (single-regional: $t$ = 0.49, $p$ = 0.621; cross-regional: $t$ = 1.14,

$p$ = 0.264) or LTC (single-regional: $t$ = −0.74, $p$ = 0.457; cross-regional: $t$ = −0.84, $p$ = 0.406). These findings further underscore the specific functional role of the EC in WM performance.

## Increased EC Connectivity with Load

Building on these findings, we hypothesized that the EC shares more information with the hippocampus and LTC under medium-to-high load conditions. To test this, we investigated whether connectivity between the EC and these regions increases with WM load. We calculated the PLV for loads 4, 6, and 8 across each channel pair connecting the regions. PLVs up to 40 Hz were computed in the time-frequency domain to capture dynamic fluctuations in functional connectivity. To assess the effect of load on PLV, we conducted repeated-measures ANOVAs with load as a within-subject factor. The results revealed a significant main effect of load on PLV between the EC and hippocampus (Fig. 5D; $F(2,24)$ = 11.95, $p$ < 0.001; post-hoc tests: load 4 vs load 6, $p$ = 0.01; load 4 vs load 8, $p$ = 0.002; load 6 vs load 8, $p$ > 0.05), as well as between the EC and LTC (Fig. 5E; $F(2,24)$ = 8.21, $p$ = 0.002;

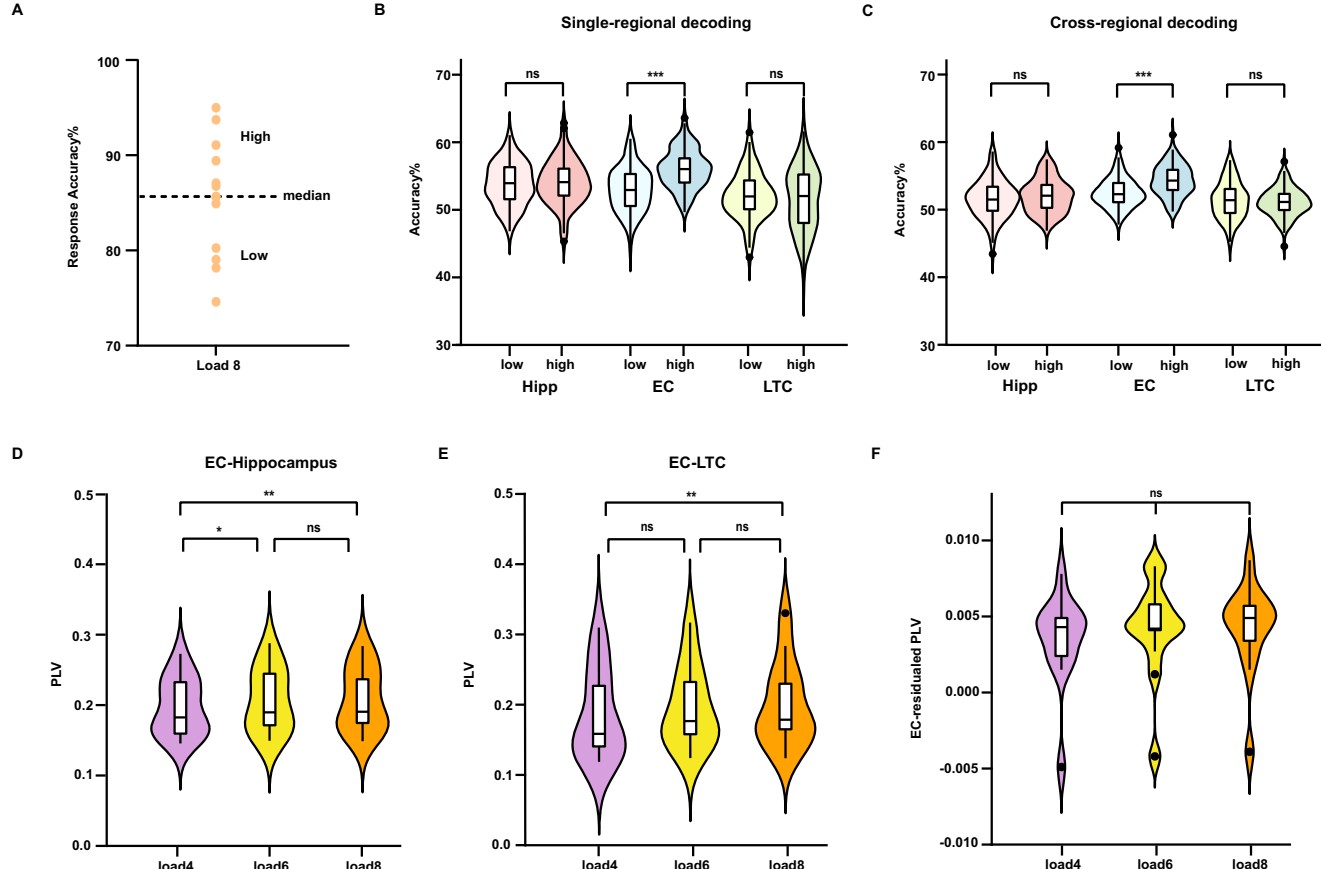

**Fig. 5 | Decoding performance of high and low behavioral groups and phase synchronization. A** Participants were divided into high-performing (7 participants) and low-performing (6 participants) groups based on median recall accuracy for load 8. **B** Single-regional decoding accuracy was significantly higher in the high-performing group (dark color) than in the low-performing group (light color) for the entorhinal cortex (EC, blue; two-sided permutation $t$ tests: $p < 0.001$), but not for the hippocampus (Hipp, red; two-sided permutation $t$ tests: $p = 0.621$) or lateral temporal cortex (LTC, green; two-sided permutation $t$ tests: $p = 0.457$) across all cross-validations ($n = 100$). ***$p < 0.001$. ( **C**) Cross-regional decoding accuracy was significantly higher in the high-performing group than in the low-performing group for the EC (two-sided permutation t-tests: $p < 0.001$), but not for the hippocampus (two-sided permutation $t$ tests: $p = 0.264$) or LTC (two-sided permutation t-tests: $p = 0.406$) across all cross-validations ($n = 100$). ***$p < 0.001$. **D** Phase locking value

(PLV) between EC and hippocampus increased significantly from load 4 (purple) to load 6 (yellow) and from load 4 to load 8 (orange) ($n = 13$ participants; repeated-measures ANOVA: load 4 vs load 6, $p = 0.01$; load 4 vs load 8, $p = 0.002$; load 6 vs load 8, $p > 0.05$). **$p < 0.01$, *$p < 0.05$. **E** PLV between EC and LTC increased significantly from load 4 to load 8 ($n = 13$ participants; repeated-measures ANOVA: load 4 vs load 6, $p = 0.10$; load 4 vs load 8, $p = 0.005$; load 6 vs load 8, $p = 0.79$). **$p < 0.01$. **F** The average EC-residualed PLV did not differ significantly across load conditions ($n = 13$ participants; repeated-measures ANOVA: all $ps > 0.05$). In the box plots shown in (**B**–**F**), the center line represents the median, and the edges of the box correspond to the lower and upper quartiles, respectively. The whiskers extend to the minimum and maximum data points at most 1.5 times the interquartile range. Source data are provided as a Source Data file.

post-hoc tests: load 4 vs load 6, $p = 0.10$; load 4 vs load 8, $p = 0.005$; load 6 vs load 8, $p = 0.79$). These results indicate a load-dependent increase in connectivity between the EC and the other two regions.

To assess whether the load-dependent connectivity increase is specific to the EC, we first examined the effect of WM load on hippocampus-LTC connectivity and observed a significant main effect ($F(2,24) = 7.26$, $p = 0.02$; post-hoc tests: load 4 vs load 6, $p = 0.12$; load 4 vs load 8, $p = 0.01$; load 6 vs load 8, $p > 0.05$). Next, we removed the EC's influence by using EC-residualized PLV as the dependent variable; under this analysis, hippocampus-LTC connectivity was no longer significantly associated with WM load (Fig. 5F; all $ps > 0.05$). These results indicate that the load-dependent increase in connectivity is specific to the EC.

## Control Analyses
### Effect of iteration count and cross-validation on decoding accuracy
To evaluate the stability of the statistical results, we also performed all statistical tests with 50 and 150 iterations/cross-validations. Our

findings were replicated with these parameters, confirming the stability of the results. The detailed results were provided in Supplementary Tables S3.

### Decoding results based on balanced trial count
To rule out the potential impact of trial count on classification performance, we randomly selected 747 trials from load 4 and load 6 to match the number of trials in load 8. This resampling procedure was repeated 10 times, and all decoding analyses were re-executed with balanced trial counts. For each resampling, statistical comparisons between brain regions were conducted separately. Our results revealed no significant differences in decoding accuracy (Supplementary Fig. S2A) or cross-region generalization (Supplementary Fig. S2C) among brain regions under low-to-medium load conditions. Under medium-to-high load conditions, the EC demonstrated the highest decoding accuracy (Supplementary Fig. S2B) and cross-region generalization (Supplementary Fig. S2D). Moreover, removing EC-shared information significantly reduced the decoding accuracy of the hippocampus and LTC (Supplementary Fig. S2E). In summary, these

results are consistent with the original findings, demonstrating that the EC exhibits enhanced decoding performance and shares load-related information with other regions under medium-to-high load conditions.

## Power and connectivity in SOZ channels

To assess the potential effects of the SOZ on our findings, we compared power and PLV between channels within and outside the SOZ, since our decoding analyses rely on power. The analysis revealed no significant differences in power between the SOZ and non-SOZ channels in the EC (linear mixed-effects model: $p = 0.24$), hippocampus ($p = 0.36$), and LTC ($p = 0.86$). Similarly, we examined the PLV between SOZ (defined as at least one SOZ channel) and non-SOZ channel pairs and found no significant differences in PLV between these channel pairs (EC-hippocampus: $p = 0.31$; EC-LTC: $p = 0.37$; hippocampus-LTC: $p = 0.09$).

## Hemispheric effects on decoding, power, and connectivity

To examine the impact of hemisphere on our findings, we performed additional analyses on the participants with bilateral electrode implantation in the hippocampus, EC, and LTC ($n = 10$). We conducted single-regional decoding analysis using the power features from the left and right hemispheres separately. The results revealed no significant differences in decoding accuracy on low-to-medium load among the three brain regions in either the left hemisphere (hippocampus: 59.06%, EC: 59.22%, LTC: 59.29%; all $ps > 0.05$) or the right hemisphere (hippocampus: 59.32%, EC: 59.39%, LTC: 59.40%; all $ps > 0.05$). Under medium-to-high load condition, decoding accuracy using EC power features was significantly higher in both the left hemisphere (hippocampus: 51.92%, EC: 54.93%, LTC: 52.46%; all $ps < 0.001$) and the right hemisphere (hippocampus: 50.72%, EC: 53.09%, LTC: 51.47%; all $ps < 0.001$). We thus did not find the hemispheric lateralization in the adaptation to WM load in these brain regions.

As a further test, we compared the power between the left and right hemispheres for each brain region, since our decoding analyses are based on power. The analysis revealed no significant hemispheric differences in power across the hippocampus ($p = 0.33$), EC ($p = 0.06$), or LTC ($p = 0.35$). Finally, we examined inter-regional connectivity between the left and right hemispheres and found no significant differences in the connectivity of EC-hippocampus ($p = 0.94$), EC-LTC ($p = 0.21$), or hippocampus-LTC ($p = 0.29$). In summary, the current study did not find the influence of hemisphere.

# Discussion

Together, the present findings highlight the EC's critical role in adapting to WM load. Specifically, the EC is more sensitive to load changes compared to the hippocampus and LTC. Cross-regional and residual-based decoding analyses indicated that the EC shares more WM-relevant information compared with the other two regions under medium-to-high load conditions. Consistent with this, the connectivity between the EC and these regions increased as cognitive demands increased. Finally, decoding performance and cross-regional generalization of the EC were strongly associated with WM performance under medium-to-high load conditions.

A key issue to address is why the EC contributes more to WM maintenance than the hippocampus or LTC under medium-to-high load conditions. One possible explanation is that the EC serves as a gateway to the hippocampus, receiving information from the neocortex and directing input to the hippocampus[19,20]. Thus, the EC acts as the initial processing station where information is processed before entering the hippocampus. Consistent with this, our cross-regional and residual-based decoding analyses suggested that the EC plays a central role in integrating and sharing information with the hippocampus and LTC, thereby enabling more efficient adaptation to increased cognitive demands. In addition, Recent rodent findings

indicated that beyond acting as an information gateway, the EC also modulates both the neocortex and hippocampus. For example, electrical stimulation targeting the EC strongly suppresses neocortical pyramidal neuron activity[21], while optogenetic perturbation of EC gamma oscillations impairs hippocampal gamma oscillations and learning processes[22]. In addition, EC neurons have been shown to modulate neocortex-hippocampus interactions in vivo[23]. Therefore, while our findings highlight the EC's integrative role in adapting to high cognitive demands, further research is needed to understand its broader modulatory functions.

Previous studies have examined the relationship between brain activation, connectivity, and varying WM loads using single-unit recordings, intracranial and scalp EEG, and fMRI. Specifically, single-unit recordings have shown that hippocampal firing during WM maintenance increases from low-to-medium load but does not continue to rise from medium-to-high load[12]. fMRI studies have observed a progressive increase in executive control network activation[6] and an inverted-U-shaped activation pattern[24] across multiple brain regions with higher WM load. In addition, electrophysiological studies have reported enhanced fronto-parietal connectivity[5] and stronger theta/alpha band hippocampus-EC connectivity[14] as WM load increases. Our study introduces an alternative approach by integrating between-region comparisons and inter-regional connectivity within a unified framework. The results suggest that the brain reallocates neural resources to meet varying cognitive demands. The EC, acting as a primary gateway between the hippocampus and neocortex and exhibiting load-dependent increases in connectivity, contributes more than the other two regions under higher cognitive demands.

The current results show that decoding accuracy on WM load, derived from power features in the hippocampus, EC, or LTC, was significantly above chance, indicating that these regions contribute to WM processing. These findings are consistent with previous reports linking WM processing to elevated activity in the hippocampus[12,25], EC[26], and LTC[27]. A resulting question is: how do these regions coordinate to support WM, and what distinct roles do they play? Despite the lack of direct evidence, we attempt to draw insights from long-term memory (LTM) research. Previous studies have primarily focused on LTM, suggesting that memory relies on circuits in the EC that connect the hippocampus with the neocortex[28]. One theory proposes that the EC processes "general" information before passing it to the hippocampus, which refines it through pattern separation and pattern completion processes[29]. Moreover, the EC recodes hippocampal outputs into a form suitable for LTM storage in the neocortex[30]. Then, during recall, a reduced memory representation stored in the hippocampus reinstates the uncompressed neocortical representations via EC connections[31]. We hypothesize that in WM, these three regions may operate via similar mechanisms. Specifically, the EC may process object-related information from the neocortex and transfer it to the hippocampus for further refinement, while hippocampal outputs are relayed back to the neocortex for further processing.

The left hemisphere of the brain is generally considered to play a dominant role in verbal memory processing[32]. As one would expect for this language-related task, a previous analysis of scalp EEG data for the same task found a clear lateralization to the left cortical hemisphere[16]. However, there was no left-hemispheric predominance in the MTL[12,16]. Since we record from patients with MTL epilepsy, we cannot rule out plastic reorganization resulting in bilateral MTL processing of this task.

It should be noted that our data were recorded in patients with epilepsy, which could affect our results. To mitigate this confounding factor, we carefully screened all trials and excluded those with signs of epileptiform activity before analysis. Still, the lack of left-hemispheric dominance might result from plastic reorganization of the MTL. However, several of our other observations seem to be independent of epilepsy. First, the low-frequency power dynamics in these regions reflect cognitive processes: they increase during encoding, remain

stable during maintenance, and decline during retrieval, consistent with previous reports[12,16,33] and are unlikely to result from epileptiform patterns[34]. Second, both power and PLV are modulated by WM load: power from all three regions successfully decoded WM load, and connectivity exhibited a load-dependent increase, effects that speak for association with cognitive brain function. Third, comparisons of power and PLV between electrodes within and outside the SOZ revealed no significant differences, indicating that this brain tissue was associated with brain function even though the tissue was subsequently resected. Together, these functional associations suggest that our findings might be generalizable to the general population.

In summary, the brain is not static but dynamically adapts to cognitive load and behavioral demands, with the EC playing a crucial role in this process. Future research incorporating widespread recordings across the brain[35] and finer subregional resolution[29,36] would contribute to a more precise understanding of how neural resource allocation facilitates adaptive responses to varying cognitive demands.

## Methods

### Participants

Thirteen epilepsy patients (six females; mean age ± SD: 36 ± 13 years) participated in this study. We listed the patients' pathology and seizure onset zone (SOZ) in Supplementary Table S1. All patients had their normal anti-seizure medication at the time of testing. Testing was performed at least 3 h before or after a seizure. Trials with signs of epileptiform activity were excluded from further analysis. Recordings were performed at the Swiss Epilepsy Center, Klinik Lengg, Switzerland. The study was approved by the local ethics committee (Kantonale Ethikkommission Zürich, PB 2016-02055), and all patients gave written informed consent. The data are freely available[37].

### Task design and behavioral metrics

We adopted a modified Sternberg task (Fig. 2A). Each trial started with a 1-second fixation screen, followed by a 2-second stimulus presentation at the center of the display, comprising a set of four, six, or eight letters. Participants were indicated to memorize these letters, with the set size defining the memory load (load 4/6/8). We define loads 4, 6, and 8 as low, medium, and high loads, respectively. After a delay (maintenance) period of 3 s, a probe letter prompted the subjects to retrieve their memory (retrieval) and to indicate by button press ("IN" or "OUT") whether or not the probe letter was a member of the letter set held in memory. Each session comprised 50 trials, approximately totaling 10 min. Each participant conducted multiple sessions across several recording days, with an average of 4.8 ± 1.9 sessions (ranging from 2 to 8).

The WM capacity of each participant was evaluated using Pashler's K, defined as $K_P = (\text{hit rate - false alarm rate}) \times N / (1 - \text{false alarm rate})$, where N is the number of letters presented. This measure is considered a reliable estimator of capacity in whole-display tasks[38]. The maximum memory capacity $K_{max}$ was defined as the highest K value observed across load conditions.

### Electrode contact localization and selection

Electrodes had 8 contacts of 1.6 mm length, with a center-to-center distance between adjacent contacts of 5 mm and a diameter of 1.3 mm (Ad-Tech, Racine, WI, www.adtechmedical.com). Channel localization was performed through post-implantation computed tomography (CT) scans and structural T1-weighted MRI scans. The CT scans were co-registered with post-implantation MRI for each patient with FieldTrip software[39], following which channels were visually identified on the merged CT-MRI images. These channel locations were subsequently projected onto the standard Montreal Neurological Institute (MNI) 152 space and assigned to specific brain regions based on the Brainnetome Atlas[40].

Electrode target regions and hemispheres varied across participants for clinical reasons. Each participant had 3-4 electrodes targeting the hippocampus and 1-2 electrodes targeting the EC. Specifically, all thirteen participants had electrodes implanted in the hippocampus of both the left and right hemispheres. Ten participants had electrodes in the EC of both hemispheres, while two participants had one electrode implanted in the left EC, and one participant had one electrode in the right EC (Supplementary Fig. S3). Across all our participants, LTC contacts were selected from the same electrode as the hippocampus and EC contacts. As a result, all thirteen participants had LTC recording sites in both the left and right hemispheres. From each electrode, we selected only one or two most medial contacts targeting the hippocampus, two most medial contacts targeting the EC, and two of three most distal contacts located centrally in the gray matter targeting the LTC[41,42]. The final dataset included a total of 91 channels situated in the hippocampus, 46 channels in the EC, and 136 channels in the LTC across all patients, averaging approximately 7.0 ± 1.2 channels per patient in the hippocampus (ranging from 6 to 8), 3.5 ± 0.8 channels per participant in the EC (with a range of 2 to 4), and 10.5 ± 2.0 channels per participant in the LTC (with a range of 6 to 12). We utilized BrainNet Viewer[43] in MATLAB (MathWorks) to visualize all the recording sites, as depicted in Fig. 2B.

### Data acquisition and preprocessing

Intracranial data were recorded using a Neuralynx ATLAS system with 4 kHz sampling rate, analog-filtered above 0.5 Hz, and downsampled offline to 1 kHz. Neural signals were filtered with 1 Hz high-pass and 200 Hz low-pass finite impulse response filters with a Hamming window, and line noise harmonics were removed via discrete Fourier transform. Each electrode contact was then re-referenced to the average signal across all contacts. The data were segmented into 4 epochs per trial: 1 s baseline during the fixation period; 2 s encoding period during the presentation of the stimulus; 3 s maintenance period; and 2 s retrieval period after probe onset. We focused on the maintenance period. We systematically inspected the raw data to identify and exclude trials with residual artifacts, including signal drift or large singular artifacts typically caused by cable movement. As a result, 65 trials were excluded for load 4 (5.5%), 36 trials for load 6 (3.9%), and 39 trials for load 8 (5.1%) across all participants. Subsequent analyses were performed on the correct trials (1054 trials for load 4, 777 trials for load 6, and 747 trials for load 8). Preprocessing routines were performed using FieldTrip[39] and customized scripts in MATLAB.

### Time-frequency analysis

Time-frequency power was computed separately for each channel in the hippocampus, EC, and LTC for each participant and correct trial. The signal was convolved with complex-valued Morlet wavelets (six cycles) to extract power information at each frequency from 1 to 100 Hz (in steps of 1 Hz) with a time resolution of 1 ms. The task-induced power was analyzed per trial using a statistical bootstrapping procedure, following previous studies[33]. Specifically, we created a null distribution by randomly selecting and averaging several data points from the baseline power (500 ms pretrial) 1000 times. The raw power at each time point during the task was then z-scored by comparing it to the null distribution to generate the z-scored power.

### Decoding analysis

We used multivariate decoding analysis to explore how the neural activity features of the hippocampus, EC, and LTC adapt to changes in cognitive demands during maintenance. The support vector machine (SVM) is widely utilized in decoding analysis in neuroimaging studies[44], particularly due to its suitability for datasets with relatively small sample sizes. Hence, our analysis was based on a linear SVM as the classifier via the LIBSVM package in MATLAB[45]. To examine the roles of various brain regions in adapting to varying

loads under fine-grained scales, we conducted binary classification using z-scored time-frequency power features, with "load 4 vs load 6" indicating the transition from low-to-medium cognitive demand and "load 6 vs load 8" indicating the transition from medium-to-high demand. To increase the impact of the analysis on a larger population[44], for each load, the power at the trial level for all participants was merged as the data (load 4: 1054 samples, load 6: 777 samples, load 8: 747 samples) used in the classification. The details of all decoding analyses were as follows:

(A) Single-regional decoding: First, we decoded WM load using power features from the hippocampus, EC, and LTC separately. Taking the decoding of "load 4 vs load 6" as an example, the power features for each trial included $40 \times 3000 = 120000$ values (where 40 denotes the frequency range from 1 to 40 Hz and 3000 denotes the time range during maintenance), which were then converted into a feature vector. Then, we split 70% of the data from each load and merged them across both loads as the training dataset. The remaining data were pooled across both loads as the testing data set. Meanwhile, to reduce the feature dimensionality, principal component analysis (PCA) was applied to the training dataset to keep several principal components (K components; $5.1 \pm 6.8$ components, ranging from 1 to 25) that explained at least 99% of the variance in the data. We also transformed the testing data set with the PCA matrix that was already fitted to the training data set. In total, we had the training data set with 70% × (load 4 + load 6) samples × K features and the testing data set with 30% × (load 4 + load 6) samples × K features. We trained the SVM classifier with a linear kernel with a cost equal to one. This procedure was replicated 100 times for the cross-validation, as was done in previous studies[46,47]. The accuracy of the classifier as a performance measures was averaged across 100 cross-validations. We performed the single-regional decoding using power features from each brain region as described above. The schematic of the single-regional decoding analysis steps is shown in Fig. 2C. In addition, we also decoded "load 6 vs load 8" using power features from the hippocampus, EC, and LTC. Aside from the differences in sample size ((load 6 + load 8) samples), all other steps were the same as described above.

(B) Cross-regional decoding analysis. To evaluate whether one brain region shares information with another, we conducted cross-regional decoding analysis. Taking the EC as an example, we first applied PCA to the EC power data and retained principal components that explained at least 99% of the variance. The transformed EC data served as the training set to decode load 4 vs load 6. Next, we applied the same PCA transformation matrix, previously fitted to the EC training data, to the hippocampus and LTC power data. The transformed hippocampus and LTC data were then separately used as test sets to evaluate the model. If the EC indeed shares information from the hippocampus or LTC, we should observe substantial decoding accuracy when using the EC-trained model to decode data from the other two regions. The schematic of the cross-regional decoding analysis steps is shown in Fig. 3A. For each brain region, we averaged the decoding accuracy achieved when testing the other two regions, resulting in the final cross-regional decoding accuracy. And we refer to each region's capacity to train on its own power features and subsequently test on the remaining two regions as generalization. We used the same method to decode load 6 vs load 8 across brain regions. In total, we conducted cross-regional decoding analysis, encompassing both load 4 vs load 6 and load 6 vs load 8 conditions, across the EC, hippocampus, and LTC.

(C) Residuals-based decoding analysis. We carried out residuals-based decoding analysis to further investigate the role of EC-

shared information in the hippocampus and LTC under medium-to-high load conditions. For each trial, we first converted the original power features across three regions to feature vectors, and used the feature vectors of hippocampus and LTC as dependent variables (y) separately, with features vectors of EC as independent variables (x), to build linear regression models, retaining the resulting residuals (120000 values) of the hippocampus and LTC as features (Fig. 4A). Then, we used the residuals after regressing out the EC from the hippocampus (Hipp$_{EC\text{-}residual}$) and LTC (LTC$_{EC\text{-}residual}$) as features to decode load 6 vs load 8. The specific method was the same as the single-regional decoding analysis, except the original power features were replaced by the residuals after removing the EC. As a control, we applied the same method to regress LTC out of hippocampus (hippocampus as dependent variables, LTC as independent variables), and conversely, hippocampus out of LTC (LTC as dependent variables, hippocampus as independent variables), using the resulting residuals in each case for decoding.

## Interregional phase synchrony

To examine whether potential interactions between brain regions increase with higher cognitive demand, we calculated the PLV. The PLV quantifies the consistency of phase relationships between electrode pairs. For each electrode pair $(a, b)$, we calculated the average phase $\varphi$ difference across trials for a given time point $t$ and frequency $f$ (as defined by equation (1)). The signal was convolved with complex Morlet wavelets (six cycles) across frequency from 1 to 40 Hz (in 1 Hz steps) with a time resolution of 1 ms. This analysis was performed for each channel pair within the same hemisphere (e.g., left EC-hippocampus, EC-LTC) during maintenance for trials of each WM load. PLV values range from 0 to 1, with values approaching 1 indicating minimal phase differences over time.

$$\mathrm{PLV}_{a,b}(t,f) = \frac{1}{N\mathrm{trials}} \left| \sum_{n=1}^{n=Ntrials} \exp\left(i\left[\varphi_{n,a}(t,f) - \varphi_{n,b}(t,f)\right]\right) \right| \quad (1)$$

## Statistical analysis

For each decoding analysis, we used a non-parametric permutation test to assess significance. Specifically, we randomly shuffled the relationship between labels and data 100 times to generate a null distribution of decoding accuracies. The true decoding accuracy was then compared to this null distribution, with values exceeding the 95th percentile ($p < 0.05$) considered significant. To compare the decoding accuracy differences between pairs of conditions, including hippocampus vs EC vs LTC, residual vs original features, and low vs high groups, we performed permutation $t$ tests.

To evaluate the statistical significance of PLVs between two loads (load 4 vs load 6, load 6 vs load 8, load 4 vs load 8), we performed repeated-measures analyses of variance (ANOVAs) with WM load as the independent variable and PLV as the dependent variable.

## Reporting summary

Further information on research design is available in the Nature Portfolio Reporting Summary linked to this article.

# Data availability

The raw data generated in this study[37] have been deposited in the public database under accession link https://doi.org/10.12751/g-node.d76994/. Source data are provided in this paper.

# Code availability

The code supporting this study is available at https://doi.org/10.5281/zenodo.15355761[48].

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

## Acknowledgements

This work received support from the following sources: National Natural Science Foundation of China (No. 32271085 to J.L. and No. 32400883 to D.C.), Beijing Natural Science Foundation (No.5244049 to D.C.), Swiss National Science Foundation (SNSF 204651 to J.S.). This manuscript was prepared with the assistance of ChatGPT to improve linguistic readability.

## Author contributions

Conceptualization, J.L. and D.C.; methodology, J.Y., J.L., and D.C.; data collection, J.S. and D.L.; patient care, L.S.; data analysis: J.Y. and D.C.; writing-original draft, J.Y., J.L., and D.C.; writing-review and editing, J.Y., C.G., J.L., and J.S.; funding acquisition, J.L., D.C., and J.S.; resources, J.S.; supervision, J.L. and D.C.

## Competing interests

The authors declare no competing interests.
