## [Transparent Peer Review file · Nature Communications]

Enhanced Role of the Entorhinal Cortex in Adapting to Increased Working Memory Load

Corresponding Author: Dr Jin Li

Version 0:

Reviewer comments:

Reviewer #1

(Remarks to the Author)

The manuscript entitled "Enhanced Role of the Entorhinal Cortex in Adapting to Increased Working Memory Load: Evidence from Intracranial EEG" by Yang and colleagues aimed to find out how the entorhinal cortex, hippocampus, and lateral temporal cortex co-operate during working memory (WM) processing, specifically when processing demands rise with higher WM loads. They used intracranial electroencephalography together with multivariate machine learning analysis to investigate activation pattern changes from low to medium and from medium to high WM loads. The analysis revealed that the change from low to medium WM load could be decoded with similar accuracy from all three regions, but the shift from medium to high load was most accurately detected from the activation patterns in the entorhinal cortex. The authors also demonstrated that the entorhinal cortex had the highest level of cross-sectional generalization and that removing its shared properties from the other regions' activation patterns reduced the decoding accuracy from them. Furthermore, decoding the change from medium to high WM loads in the entorhinal cortex was more accurate with individuals with high WM performance.

As a general comment, we find the study highly interesting and the combination of this WM paradigm and intracranial EEG recordings of the hippocampal system has novelty. The methods are well selected to address this research question, and the findings are important. Regarding the evaluation, our expertise covers WM and related brain mechanisms. However, we are not specialists in intracranial EEG methods and recommend another reviewer to evaluate that part from the experimental setup and data processing viewpoint. Please find below our comments that hopefully help improving the manuscript quality prior to publication.

Comments section by section:

Abstract

Authors use the concept decoding accuracy in several sentences, but it remains a bit unclear from the text what information was decoded (=comparisons).

The findings of decoding analyses using EC data are described in detail while the connectivity results as well as concluding remarks are very shortly mentioned and not clearly connected to the main findings. The authors could consider if the results text could be expressed in a more balanced and integrated way and if the conclusions could be a bit more specific.

Introduction

When discussing previous studies focusing on neural load effects of WM processing, the authors state that (lines 74-76): "Furthermore, these findings highlight the limitations of focusing solely on activity within individual brain regions, indicating the necessity of considering WM load regulation across multiple brain regions and their interactions." We do not fully follow the reasoning behind this sentence, given that the cited studies (for example Wang et al. 2024; Zhao et al. 2024; Van Snellenberg et al. 2015) have not concentrated on individual brain regions but rather have had a more systems-level focus on working memory. Some clarification is needed here.

Overall, WM is a rather broad concept that consists of various component processes, some of which can be studied with

Sternberg task that was used in the present study. Those specific cognitive processes (encoding, maintenance, and retrieval) that concern this manuscript are mentioned in the methods, but maybe a sentence that would specify what aspect of WM was studied (and connecting that more closely to the relevant neuroscience literature) already in the introduction would help the general audience to get a better idea of the research context.

The last text paragraph in the introduction and Figure 1 legend have overlapping information. Streamlining the text here by removing some bits from the manuscript body that repeat the contents of the figure legend, and perhaps putting a more general citation to the figure legend and expanding that part a bit, might help the reader to follow the story.

Methods

Description of the clinical characteristics of the study population is lacking. I understand that the research question does not relate to epilepsy, but at least theoretically the medical history could influence the observed results. Do the patients have intact cognitive functions and has epilepsy possibly affected their brain structure and function? Instead of electrode placements (which could be moved to later section that is about the electrodes), maybe the Participants section could include a bit more information regarding the sample that would help interpreting the reproducibility of the findings.

Check the consistency related to instances referring to specific load level (e.g. load 1 vs. load1 vs. load size 1, sometimes vs is also written without the dot and sometimes with the dot).

"The maintenance of WM information elicited z-scored power changes in the 1-40 Hz frequency range." Is this sentence needed in the Methods section should it rather be in the Results section?

Lines 203-204: "Subsequent decoding analyses were conducted exclusively based on correct trials." This information was already given in the data acquisition and preprocessing paragraph, remove from here?

There may be a mistake in line 245: "The schematic of the single-decoding analysis steps is shown in Fig 3A". Should this instead be "The schematic of the cross-regional analysis...?"

Lines 222-223: "...that explained 99% of the variance in the data" Do the authors mean ...that explained at least 99% of the variance in the data (or >99%)? Probably not 99% exactly.

How were the number of the iterations and cross-validations selected for the statistical testing and was the stability of the values examined? This seems like a small number.

Please correct the typo in line 256: "...medium-to-high load= condition" -> medium-to-high load condition

Results

One concern with the results is that the overall decoding accuracies of load level changes seem rather modest in all three inspected areas and both in low-to-medium and medium-to-high conditions, the highest accuracy being around 56 %. Did the authors test whether the decoding accuracy was statistically different from the chance-level? How do the decoding accuracies in this study relate to findings from previous studies? This point is also important considering that in the discussion section the authors state that "The current results are consistent with previous research, indicating that the hippocampus, LTC, and EC all contribute to WM processing." It is somewhat difficult to find evidence for the role of hippocampus and LTC in WM processing from the current results, also given that the analysis comparing behavioral results were only conducted on the entorhinal cortex (see also our other point below).

An important aspect of the current research was the comparison of the power features between high- and low-performing individuals. Why were these analyses only conducted on entorhinal cortex, and not on hippocampus or lateral temporal cortex? Would those areas show similar associations with behavioral performance? As decoding is a very powerful approach, this type of cross-validation would be helpful to understand the specificity of the reported effects.

Similarly, why were the phase-locking value connectivity analyses only conducted between the entorhinal cortex and other areas, and not between hippocampus and lateral temporal cortex? It would be interesting to see whether the load-dependent connectivity increase is specific to the entorhinal cortex or something generally seen between all inspected areas.

In line 317 the sentence starts And... Please correct.

In line 323 the abbreviation SVM is defined for the second time.

Lines 337-338: "Across all participants, The median memory capacity across all participants..." Please correct.

Figure 2A left. The distributions of the data do not look very nice. Could the authors comment on that.

Line 540: "Result, Results showed significant..." Please correct.

Overall, in the results section, quite a bit of text is devoted to summaries of the key findings that already contain a bit of interpretation (end of each result section). Maybe extensive summaries with interpretation are not needed if the results are

described in a clear way from the beginning.

Discussion

The sentence in lines 551-553 "Cross-regional and residual-based decoding analyses indicated that the EC shares more WM relevant information with the other two regions under medium-to-high load" could easily be interpreted so that EC shares more information under medium-to-high load than under low-to-medium load, opposed to that EC shares more information compared with the other regions as it probably should say based on the results. Some clarification could be helpful here.

In lines 558-559 it is said that "A key issue to address is why the EC, but not the hippocampus or LTC, contributes to the active maintenance of medium-to-high load?" Consistent with our previous point (see the comments for the results section), it was not clear whether the role of the hippocampus or LTC in WM maintenance was actually statistically tested. Is it then correct to say that they do not contribute to WM maintenance in medium-to-high loads?

Some previous WM load studies are presented in the third paragraph of the discussion (line 576 onwards). From there, one could be left with the impression that not many studies exist where WM load effects were studied beyond local activation patterns. This can feel a bit misleading, considering also that the authors have cited several system-level and/or connectivity studies about WM load effects (including e.g. Wang et al. 2024; Zhao et al. 2024; Van Snellenberg et al. 2015 - see also our previous comment about the introduction) and that previous studies, presumably from the same dataset as the current study, have shown WM load effects in the connectivity between the hippocampus and entorhinal cortex (Li et al. 2024) and between the hippocampus and cortex (Boran et al. 2019). Both studies are cited in the manuscript. The reader could benefit from a more elaborated discussion of these previous studies, including the studies conducted previously from this same dataset.

Finally, the language could be improved, and typos carefully checked throughout the manuscript.

Reporting summary

This report states shortly that no data were excluded, but the manuscript describes trial exclusion that was based on the visual inspection. Please check the consistency, expand how the visual inspection was done, and include information of the final number of included (correct) trials to the manuscript.

Reviewer #2

(Remarks to the Author)

In this manuscript, the authors seek to test how the brain flexibly adapts to different levels of working memory (WM) load. They conducted a Sternberg working memory task in patients with intracranial electrodes and specifically investigated modulation of spectral signals in entorhinal cortex (EC), hippocampus, and lateral temporal cortex (LTC) in response to low (4 letters), medium (6), and high (8) WM loads. The authors report a series of complimentary analyses demonstrating a specific contribution of EC to WM maintenance under the high (relative to medium) load condition. Signal from all three regions can be used to decode low vs. medium and medium vs. high load conditions; however, when overlapping data from EC is regressed out of the hippocampal and LTC data, decoding accuracy is significantly reduced in those regions. Cross-region decoding indicates that there is shared information between EC and hippocampus and between EC and LTC. EC decoding accuracy is higher in participants with high compared to low performance. Phase locking values between EC and hippocampus and between EC and LTC are significantly modulated by WM load. The authors conclude that the EC may serve WM in the same manner as it does in LTM, namely by processing general information prior to sending this information to the hippocampus and relaying hippocampal outputs to the cortex for storage and retrieval.

Overall, this is a well written paper with robust and compelling findings. Each analysis provides important and complimentary insights into the role of EC in WM maintenance which will be of interest to the field. I have a few comments and points of clarification which I outline below.

Major

1. As expected, the number of trials differs across load condition (Lines 210-211, "load 4: 1054 samples, load 6: 777 samples, load 8: 747 samples"); however, it is not clear from the methods whether measures were taken to address this imbalance in the classification procedure. The methods mention that 70% of trials were used as training data, but the trial counts may still be imbalanced which can bias classification performance.
2. It is not always clear which features are used for classification. Are the PCA features only used for the within-region classification analysis? The cross-region analysis references "power features" which implies that non-PCA features were used which seems potentially in danger of over fitting given the very large number of features.
3. The authors state that, "Our findings support the notion that increased WM demands require a more integrated, less modular network configuration" (lines 582-583) and reference "the WM network" (line 607). These statements seem to be a bit of an over-interpretation of the data given that the authors did not directly measure modularity and the regions considered here do not encompass the entirety of the regions recruited during WM maintenance (and thus cannot be considered "the WM network").
4. The authors show that EC generalization outperforms hippocampal generalization (Lines 400-401), but is the EC

generalization performance itself significantly above chance? Relatedly, do the LTC-residualized hippocampus and hippocampal-residualized LTC classifiers perform above chance?

5. Does the PLV between hippocampus and LTC vary as a function of load? (It might be informative to perform the phase synchrony analysis on the EC-residualized data.)

6. Can the authors clarify this seeming discrepancy: on lines 157-159, the authors state that "Targeted regions ... included .. the EC in the left (n = 12) and right (n = 11) hemisphere" which seems to imply that one of the 13 participants did not have any EC electrodes. However, on lines 165-166, the authors state that there were " 3.5 ± 0.8 channels per participant in the EC (with a range of 2 to 4)."

Minor

1. In general, the level of detail around numbers of trials, electrodes, etc. is excellent and the authors provide means, minimums, and maximums. The same should be provided for the number of sessions (p 6) and the number of principal components (line 222) as currently the authors state, "with some participants undertaking multiple sessions, up to a maximum of eight" and that there were "several principal components."

2. I found lines 101-103 "Our findings show..." a little confusing because this statement references figure 1B which does not show data but a schematic/predicted results.

3. The authors state on lines 129-130, "We adopted a modified Sternberg task that temporally dissociated the processes of WM encoding, maintenance, and retrieval." I found the 'temporal dissociation...' phrase confusing, don't all Sternberg (and all WM tasks) temporally dissociate study/maintenance/test?

4. In a few cases some of the terminology was a bit underspecified. For instance, the authors state that they used "consistent parameters for time-frequency analysis" (lines 275-276), but it's unclear what "consistent" means in this context.

Reviewer #3

(Remarks to the Author)

This is a very interesting manuscript evaluating entorhinal cortex, hippocampal, and lateral temporal cortex activity during varying working memory tasks in patients with drug-resistant epilepsy. The manuscript is well communicated with valid, relevant methods and robust interpretation of results. While this study has great strengths, the authors don't mention the limitations of studying memory function in patients with drug-resistant epilepsy. In addition, there is information that is missing, mostly in the Methods section, which would greatly improve the applicability and generalizability of the manuscript that are listed below.

There is no information regarding the patient's clinical features, including the following:

1. Side of recording (did all patients have bilateral recordings?)
2. MRI findings (i.e. hippocampal sclerosis)
3. Pathology findings
4. Medications for each patient at the time of testing

Since memory has lateralizing features (i.e. the left hemisphere is associated with episodic verbal memory), the authors should compare how the side of the recording affects WM performance or at least add side as a covariate.

Information should be given on the epileptogenicity of all recording sites (i.e. frequency of epileptogenic discharges and seizure activity/seizure onset zone -if identified).

The authors should discuss the shortcomings of studying electrical activity in this patient population and its applicability in the general population.

Reviewer #4

(Remarks to the Author)

Version 1:

Reviewer comments:

Reviewer #1

(Remarks to the Author)

Authors have carefully addressed all my concerns.

(Remarks on code availability)

Reviewer #2

(Remarks to the Author)

The authors have addressed the majority of my concerns. My only remaining point is that the balanced classification analysis currently described in the supplementary should be moved to the main text and replace the imbalanced classification analysis, as it is not valid to conduct classification when classes are imbalanced.

(Remarks on code availability)

Reviewer #3

(Remarks to the Author)

The authors fully addressed concerns in the revision.

(Remarks on code availability)

I have reviewed the code in the link above, and it appears to be suitable for its intended purpose. I have not installed and run the code.

Reviewer #4

(Remarks to the Author)

(Remarks on code availability)

Reviewer comments

We thank all expert Reviewers for their encouraging and constructive comments, which have undoubtedly improved the quality of our manuscript. Here, we provide a point-by-point response to all comments. Our responses are in blue and new text in red.

1 Reviewer #1:

The manuscript entitled "Enhanced Role of the Entorhinal Cortex in Adapting to Increased Working Memory Load: Evidence from Intracranial EEG" by Yang and colleagues aimed to find out how the entorhinal cortex, hippocampus, and lateral temporal cortex co-operate during working memory (WM) processing, specifically when processing demands rise with higher WM loads. They used intracranial electroencephalography together with multivariate machine learning analysis to investigate activation pattern changes from low to medium and from medium to high WM loads. The analysis revealed that the change from low to medium WM load could be decoded with similar accuracy from all three regions, but the shift from medium to high load was most accurately detected from the activation patterns in the entorhinal cortex. The authors also demonstrated that the entorhinal cortex had the highest level of cross-sectional generalization and that removing its shared properties from the other regions' activation patterns reduced the decoding accuracy from them. Furthermore, decoding the change from medium to high WM loads in the entorhinal cortex was more accurate with individuals with high WM performance.

As a general comment, we find the study highly interesting and the combination of this WM paradigm and intracranial EEG recordings of the hippocampal system has novelty. The methods are well selected to address this research question, and the findings are important. Regarding the evaluation, our expertise covers WM and related brain mechanisms. However, we are not specialists in intracranial EEG methods and recommend another reviewer to evaluate that part from the experimental setup and data processing viewpoint. Please find below our comments that hopefully help improving the manuscript quality prior to publication.

Reply: We thank the reviewer for this appreciation.

1.1 Abstract

1.1.1 Concept of decoding accuracy

Authors use the concept decoding accuracy in several sentences, but it remains a bit unclear from the text what information was decoded (=comparisons).

Action: We are now more clearly illustrating what information was decoded in the **Abstract**. It now reads:

... Using multivariate machine learning analysis, we decoded WM load using the power from each region as neural features. ...

1.1.2 Description of results and conclusions

The findings of decoding analyses using EC data are described in detail while the connectivity results as well as concluding remarks are very shortly mentioned and not clearly connected to the main findings. The authors could consider if the results text could be expressed in a more balanced and integrated way and if the conclusions could be a bit more specific.

Reply: We thank the reviewer for this valuable suggestion.

Action: We have now added a description of the connectivity results and provided a more specific conclusion:

... The results showed that the EC exhibited both higher decoding accuracy on medium-to-high load and superior cross-regional generalization. Further analysis revealed that removing EC-related information significantly reduced residual decoding accuracy in the hippocampus and LTC. Additionally, we found that WM maintenance was associated with enhanced phase synchronization between the EC and other regions. This inter-regional communication increased as WM load rose. These results suggest that under higher WM load, the brain relies more on the EC, a key connector that links and shares information with the hippocampus and LTC.

1.2 Introduction

1.2.1 Statement on previous research

When discussing previous studies focusing on neural load effects of WM processing, the authors state that (lines 74-76): "Furthermore, these findings highlight the limitations of focusing solely on activity within individual brain regions, indicating the necessity of considering WM load regulation across multiple brain regions and their

interactions." We do not fully follow the reasoning behind this sentence, given that the cited studies (for example Wang et al. 2024; Zhao et al. 2024; Van Snellenberg et al. 2015) have not concentrated on individual brain regions but rather have had a more systems-level focus on working memory. Some clarification is needed here.

Reply: We thank the reviewer for the valuable comment and apologize for any misunderstanding caused by our previous wording. Our aim was not to imply that previous studies examined only single-region effects. Although studies such as Wang et al. (2024), Zhao et al. (2024), and Van Snellenberg et al. (2015) analyzed WM load effects across multiple regions, they did so by evaluating each region's activity separately. In contrast, our study directly compared the contributions of different regions and explored how these contributions change with increasing cognitive demand, emphasizing the EC's role as a key connector linking the hippocampus and LTC. Then, our comprehensive approach integrates cross-regional decoding generalization, residual-based decoding by removing EC-shared activity, and assessment of load-dependent inter-regional connectivity. We believe this approach confirms our hypothesis and provides a more nuanced view than previous studies.

Action: We have revised the manuscript to more clearly explain our point. It now reads:

... Furthermore, previous studies have primarily focused on the relationship between neural activity in individual brain regions and WM load, examining each region separately. Here, we examine how the contributions of different brain regions vary with increasing load and assess the role of inter-regional connectivity.

1.2.2 Specific cognitive processes

Overall, WM is a rather broad concept that consists of various component processes, some of which can be studied with Sternberg task that was used in the present study. Those specific cognitive processes (encoding, maintenance, and retrieval) that concern this manuscript are mentioned in the methods, but maybe a sentence that would specify what aspect of WM was studied (and connecting that more closely to the relevant neuroscience literature) already in the introduction would help the general audience to get a better idea of the research context.

Reply: We fully agree with the Reviewer's suggestion to clarify the specific aspect of WM being studied. Our research focuses on WM maintenance.

Action: We have now specified the cognitive processes we investigate in the **Introduction** section:

... leaving open questions about how these regions coordinate to **maintain the increased WM content**.

1.2.3 Text description of Figure 1

The last text paragraph in the introduction and Figure 1 legend have overlapping information. Streamlining the text here by removing some bits from the manuscript body that repeat the contents of the figure legend, and perhaps putting a more general citation to the figure legend and expanding that part a bit, might help the reader to follow the story.

Reply: We thank the reviewer for this helpful suggestion.

Action: We have revised the last paragraph of the **Introduction** and the **Figure 1** legend to eliminate overlapping information:

... We investigated neural adaptations to WM load by machine learning decoding analyses under low-to-medium and medium-to-high load conditions separately. Decoding analyses were conducted using single-regional power features (**Fig. 1B**), cross-regional decoding generalization (**Fig. 1C**), and residual-based decoding to remove EC-shared activity (**Fig. 1D**). Additionally, functional connectivity analysis (**Fig. 1E**) was used to assess load-dependent changes in inter-regional connectivity. Our findings indicate that the EC exhibits superior decoding performance on medium-to-high WM load, shares load-related information with other regions, and displays load-dependent increases in its functional connectivity with the hippocampus and LTC.

Fig. 1 Overview of the analysis pipeline. (A) Schematic illustration of the WM task and iEEG recording sites in the EC, hippocampus, and LTC. (B) Under medium-to-high load condition, decoding accuracy based on EC power features was higher than that derived from the hippocampus or LTC. (C) Cross-regional decoding, in which decoders trained on one region's data were tested on another, revealed that EC-based decoders demonstrated the highest generalization under medium-to-high load condition. (D) Residual decoding analysis showed that removing neural activity shared with the EC significantly reduced decoding accuracy in the hippocampus and LTC under medium-to-high load condition. (E) Functional connectivity analysis indicated that the phase locking value (PLV) between the EC and other regions increased with

enhanced WM load. The brain (A, C) was visualized by BrainNet Viewer toolbox (www.nitrc.org/projects/bnv/).

1.3 Methods

1.3.1 Clinical characteristics and sample details

Description of the clinical characteristics of the study population is lacking. I understand that the research question does not relate to epilepsy, but at least theoretically the medical history could influence the observed results. Do the patients have intact cognitive functions and has epilepsy possibly affected their brain structure and function? Instead of electrode placements (which could be moved to later section that is about the electrodes), maybe the Participants section could include a bit more information regarding the sample that would help interpreting the reproducibility of the findings.

Reply: We thank the reviewer for the valuable suggestion. We have now included information on the patients' pathology in **Supplementary Table S1**. All patients had their normal anti-seizure medication at the time of testing. While some patients had cognitive deficits, all patients scored well in the task (median 92.5% correct trials). Furthermore, to assess whether epilepsy might have influenced our findings, we compared power and phase-locking values between electrodes within and outside the seizure onset zone (SOZ), since our decoding analyses rely on power. The results showed no significant differences between SOZ and non-SOZ channels, our findings seem to be independent of the SOZ-related effects.

Action: Following the Reviewer's comment, we now report in the **Methods** section: **Participants**. ... We listed the patients' pathology and seizure onset zone (SOZ) in **Supplementary Table S1**. All patients had their normal anti-seizure medication at the time of testing. Testing was performed at least 3 hours before or after a seizure. Trials with signs of epileptiform activity were excluded from further analysis. ...

And in the **Results** section:

... The average memory accuracy across all participants was $92.04 \pm 3.35\%$ (median 92.5% correct trials), indicating that all patients scored well in the task. ...

We have described the results of the comparison between SOZ and non-SOZ channels in the **Results** section:

Power and Connectivity in SOZ Channels

To assess the potential effects of the SOZ on our findings, we compared power and PLV between channels within and outside the SOZ, since our decoding analyses rely on power. The analysis revealed no significant differences in power between the SOZ and non-SOZ channels in the EC (linear mixed-effects model: $p = 0.24$), hippocampus ($p = 0.36$), and LTC ($p = 0.86$). Similarly, we examined the PLV between SOZ (defined as at least one SOZ channel) and non-SOZ channel pairs and found no significant differences in PLV between these channel pairs (EC-hippocampus: $p = 0.31$; EC-LTC: $p = 0.37$; hippocampus-LTC: $p = 0.09$).

Regarding the electrode information, we have moved it to the **Electrode Contact Localization and Selection in Methods** section:

Electrode contact localization and selection. Electrodes had 8 contacts of 1.6 mm length, with a center-to-center distance between adjacent contacts of 5 mm and a diameter of 1.3 mm (Ad-Tech, Racine, WI, www.adtechmedical.com). ...

And we have now included information on patients' pathology in the **Supplementary Table S1**:

Supplementary Table S1. Subject characteristics.

Patient	Age [y]	Sex	Pathology	Seizure onset zone	
1	24	f	Xanthoastrozytoma WHO II	Hippocampus	right
2	39	m	Hippocampal gliosis	Hippocampus	right
3	18	f	Hippocampal sclerosis	Hippocampus	left
4	28	m	Posttraumatical lesion	Hippocampus	bilateral
5	31	m	Hippocampal sclerosis	Amygdala	right
6	47	m	Hippocampal sclerosis	Hippocampus	right
7	56	f	mesial temporal sclerosis	Entorhinal cortex	right
8	19	f	mesial temporal sclerosis	Amygdala	right
9	35	m	non-lesional	Hippocampus	bilateral
10	51	f	Hippocampal sclerosis	Hippocampus	left
11	30	m	Dysembryoplastic neuroepithelial tumour	Hippocampus	left
12	29	f	non-lesional	Hippocampus	bilateral

1.3.2 Description of load level

Check the consistency related to instances referring to specific load level (e.g. load 1 vs. load1 vs. load size 1, sometimes vs is also written without the dot and sometimes with the dot).

Reply: We appreciate the reviewer's careful checks and apologize for our carelessness.

Action: Following the reviewer's comment, we have carefully revised the manuscript and thoroughly reviewed the text to ensure consistent usage of load levels. Specifically, we have removed the unnecessary "size" and standardized the terminology to "load 4," "load 6," and "load 8." Additionally, we have standardized the use of "load 4 vs load 6" and "load 6 vs load 8" without the dot in "vs".

1.3.3 Time-frequency analysis results

"The maintenance of WM information elicited z-scored power changes in the 1-40 Hz frequency range." Is this sentence needed in the Methods section should it rather be in the Results section?

Action: We have moved this sentence to the **Results** section:

Greater Load Sensitivity of EC under Medium-to-High Demands

... The maintenance of WM information elicited z-scored power changes in the 1-40 Hz frequency range. Accordingly, neural activity within this range was included in the subsequent multivariate decoding analyses.

1.3.4 Information removal

Lines 203-204: "Subsequent decoding analyses were conducted exclusively based on correct trials." This information was already given in the data acquisition and preprocessing paragraph, remove from here?

Action: We have removed this sentence.

1.3.5 Schematic of cross-regional analysis

There may be a mistake in line 245: "The schematic of the single-decoding analysis steps is shown in Fig 3A". Should this instead be "The schematic of the cross-regional analysis..."?

Action: We apologize for this typographical error. We have revised the sentence and carefully reviewed the entire manuscript to avoid similar mistakes. The revised text now reads:

... The schematic of the **cross-regional** decoding analysis steps is shown in **Fig. 3A**. ...

1.3.6 Variance explanation percentage

Lines 222-223: "...that explained 99% of the variance in the data" Do the authors mean ...that explained at least 99% of the variance in the data (or >99%)? Probably not 99% exactly

Action: We have revised this sentence:

... that explained **at least** 99% of the variance in the data. ...

1.3.7 Iterations and cross-validations

How were the number of the iterations and cross-validations selected for the statistical testing and was the stability of the values examined? This seems like a small number.

Reply: We fully agree that confirming the number of iterations and cross-validations does not affect the decoding results is important. In the current study, the number of iterations and cross-validations was determined based on based on prior research ^{1, 2}, with both set to 100. In addition to our original setting of 100 iterations and 100 cross-validations, we reran the decoding analyses using 50 and 150 iterations/cross-validations, respectively. Our findings were replicated with these new parameters, confirming the stability of the results.

Action: We have now added the description and results of decoding analyses using 50 and 150 iterations/cross-validations in the **Results** section:

Control Analyses

Effect of Iteration Count and Cross-Validation on Decoding Accuracy

To evaluate the stability of the statistical results, we also performed all statistical

tests with 50 and 150 iterations/cross-validations. Our findings were replicated with these new parameters, confirming the stability of the results. The detailed results were provided in **Supplementary Tables S3**.

And in **Supplementary Table S3**:

Supplementary Table S3. Decoding accuracy with 50, 100, and 150 iterations and cross-validations. Each decoding result presented the mean accuracy with the corresponding chance level (mean accuracy \pm SD [chance level]).

			50 times	100 times	150 times
Single-regional (%)	load 4 vs load 6	EC	56.45 \pm 1.14 [52.88]	56.44 \pm 1.10 [53.16]	56.35 \pm 1.29 [52.87]
		Hippocampus	56.34 \pm 3.84 [52.26]	56.33 \pm 3.94 [52.37]	56.25 \pm 4.42 [50.32]
		LTC	56.17 \pm 3.69 [50.45]	56.08 \pm 3.84 [50.37]	56.16 \pm 3.92 [52.34]
	load 6 vs load 8	EC	55.43 \pm 1.83 [51.05]	55.49 \pm 1.72 [51.05]	55.62 \pm 1.75 [51.04]
		Hippocampus	52.85 \pm 1.70 [50.95]	52.84 \pm 1.67 [50.73]	52.89 \pm 2.28 [50.70]
		LTC	54.00 \pm 2.29 [51.02]	53.82 \pm 2.10 [51.02]	53.81 \pm 1.91 [51.01]
Cross-regional (%)	load 4 vs load 6	EC	57.14 \pm 1.24 [52.96]	57.25 \pm 1.14 [53.27]	57.21 \pm 1.26 [52.63]
		Hippocampus	57.06 \pm 1.63 [50.68]	57.14 \pm 1.83 [50.42]	57.19 \pm 0.41 [50.37]
		LTC	57.65 \pm 1.26 [57.30]	57.51 \pm 1.47 [57.25]	57.50 \pm 1.02 [57.21]
	load 6 vs load 8	EC	53.07 \pm 2.63 [50.15]	53.13 \pm 2.61 [50.07]	53.07 \pm 1.25 [49.99]
		Hippocampus	50.47 \pm 2.12 [50.29]	50.45 \pm 2.25 [50.21]	50.61 \pm 1.57 [50.16]
		LTC	51.89 \pm 1.75 [50.71]	51.97 \pm 1.97 [50.63]	51.96 \pm 1.37 [50.67]
Residuals-based (%)	EC-residual	Hippocampus	50.38 \pm 1.40 [50.11]	50.34 \pm 1.33 [50.02]	50.34 \pm 1.14 [49.99]
		LTC	50.40 \pm 1.1 [50.16]	50.31 \pm 1.41 [50.11]	50.42 \pm 1.56 [50.37]
	Control analysis	Hippocampus	51.30 \pm 0.94 [50.38]	51.36 \pm 1.22 [50.31]	51.29 \pm 1.08 [50.26]
		LTC	50.96 \pm 1.52 [50.05]	50.78 \pm 1.48 [50.05]	51.70 \pm 1.47 [50.76]
Single-regional groups (%)	low group	EC	53.19 \pm 3.00 [50.26]	53.21 \pm 3.24 [50.24]	53.37 \pm 3.19 [50.26]
		Hippocampus	53.91 \pm 3.20 [51.18]	53.89 \pm 3.16 [50.98]	54.08 \pm 3.08 [50.16]
		LTC	52.20 \pm 3.37 [50.67]	52.18 \pm 3.47 [50.57]	52.05 \pm 3.18 [50.58]
	high group	EC	56.24 \pm 2.67 [50.25]	56.22 \pm 2.87 [50.13]	56.31 \pm 2.68 [50.11]

		Hippocampus	54.04 ± 3.75 [50.30]	54.12 ± 3.32 [50.20]	54.16 ± 3.43 [50.97]
		LTC	51.74 ± 4.66 [50.32]	51.76 ± 4.60 [50.18]	51.64 ± 4.39 [50.15]
Cross-regional groups (%)	low group	EC	52.64 ± 2.24 [50.42]	52.57 ± 2.17 [50.29]	52.62 ± 2.09 [50.22]
		Hippocampus	51.43 ± 2.96 [50.24]	51.53 ± 2.81 [50.16]	51.49 ± 2.18 [50.25]
		LTC	51.58 ± 2.58 [50.62]	51.64 ± 2.72 [50.48]	51.57 ± 2.15 [50.42]
	high group	EC	54.40 ± 2.14 [50.19]	54.39 ± 2.33 [50.12]	54.42 ± 1.99 [50.10]
		Hippocampus	51.89 ± 2.27 [50.14]	51.95 ± 2.46 [50.12]	51.85 ± 2.02 [50.12]
		LTC	51.39 ± 2.24 [50.24]	51.36 ± 2.13 [50.14]	51.32 ± 2.09 [50.16]

1.3.8 Sentence correction

Please correct the typo in line 256: "...medium-to-high load= condition" -> medium-to-high load condition

Reply: We thank the reviewer for pointing out this typo.

Action: We have corrected the typo to "medium-to-high load condition".

1.4 Results

1.4.1 Comparison of decoding accuracy to chance-level

One concern with the results is that the overall decoding accuracies of load level changes seem rather modest in all three inspected areas and both in low-to-medium and medium-to-high conditions, the highest accuracy being around 56 %. Did the authors test whether the decoding accuracy was statistically different from the chance-level?

Reply: We thank the reviewer for this important suggestion. We agree that assessing whether the decoding accuracy is statistically above chance level is crucial. In the current study, significance was assessed by comparing the observed decoding accuracy to a null distribution, with an average accuracy exceeding the 95% threshold considered significant. Our analyses confirmed that all decoding results in the present study were significantly above chance level.

Action: Following the Reviewer's comment, we tested whether the decoding accuracies exceeded chance level, and all our analyses showed statistically

significant results above chance, confirming the robustness of our findings. We have added the corresponding result descriptions to the **Results** section.

In **Greater Load Sensitivity of EC under Medium-to-High Demands:**

... To assess the statistical significance of the decoding results, we created a null distribution of the decoding accuracy by shuffling the relationship between labels and data 100 times. Decoding accuracy exceeding the 95% threshold of this null distribution was considered statistically significant. The results showed that the decoding accuracies on low-to-medium and medium-to-high load conditions using power features from the hippocampus, EC, and LTC were significantly above chance level. The detailed 95% threshold values are provided in **Supplementary Table S2**. ...

and in **Higher Cross-Regional Generalization of EC under Medium-to-High Demands:**

The cross-regional decoding accuracies on low-to-medium and medium-to-high load conditions across hippocampus, EC, and LTC were significantly above chance level (see **Supplementary Table S2** for details). ...

and in **Significant Decoding Accuracy Reduction after Removing EC Information:**

... the residual-based decoding accuracies were significantly higher than chance level for the hippocampus ($50.34 \pm 1.33\%$) and the LTC ($50.31 \pm 1.41\%$; see **Supplementary Table S2** for details). ...

... The analysis showed that decoding accuracies remained significantly above chance level for LTC-residualized ($51.36 \pm 1.22\%$) in the hippocampus and the Hipp-residualized ($50.78 \pm 1.48\%$; **Supplementary Table S2**) in the LTC. ...

and in **EC Activity Pattern Linked to Better Memory Performance**

The single-regional and cross-regional decoding accuracies were significantly above chance level in both the high- and low-performing groups for the EC (low-group: single-regional $53.21 \pm 3.24\%$, cross-regional $52.57 \pm 2.17\%$; high-group: single-regional $56.22 \pm 2.87\%$, cross-regional $54.39 \pm 2.33\%$), hippocampus (low-group: single-regional $53.89 \pm 3.16\%$, cross-regional $51.53 \pm 2.81\%$; high-group: single-

regional $54.12 \pm 3.32\%$, cross-regional $51.95 \pm 2.46\%$), and LTC (low-group: single-regional $52.18 \pm 3.47\%$, cross-regional $51.64 \pm 2.72\%$; high-group: single-regional $51.76 \pm 4.60\%$, cross-regional $51.36 \pm 2.13\%$; **Supplementary Table S2**). ...

and in **Supplementary Table S2**:

Supplementary Table S2. Chance level for all decoding analyses.

		EC	Hippocampus	LTC
Single-regional (%)	load 4 vs load 6	53.16	52.37	50.37
	load 6 vs load 8	51.05	50.73	51.02
Cross-regional (%)	load 4 vs load 6	53.27	50.42	57.25
	load 6 vs load 8	50.07	50.21	50.63
Residuals-based (%)	EC-residual	/	50.02	50.11
	Control analysis	/	50.31	50.05
Single-regional groups (%)	low group	50.24	50.98	50.57
	high group	50.13	50.20	50.18
Cross-regional groups (%)	low group	50.29	50.16	50.48
	high group	50.12	50.12	50.14

1.4.2 Relation of current decoding accuracy to previous studies

How do the decoding accuracies in this study relate to findings from previous studies? This point is also important considering that in the discussion section the authors state that "The current results are consistent with previous research, indicating that the hippocampus, LTC, and EC all contribute to WM processing." It is somewhat difficult to find evidence for the role of hippocampus and LTC in WM processing from the current results, also given that the analysis comparing

behavioral results were only conducted on the entorhinal cortex (see also our other point below).

Reply: We thank the Reviewer for raising this important point, which allows us to clarify the roles of the hippocampus and LTC in WM processing.

First, our results demonstrate that decoding accuracies for WM load using power features in both the hippocampus and LTC were significantly above chance, indicating that these regions do contribute to WM processing. However, under medium-to-high load condition, their decoding accuracies were lower than those of the EC, suggesting a comparatively smaller contribution under increased load. Furthermore, comparisons between high- and low-performing individuals showed no significant differences in decoding accuracy or cross-regional generalization for the hippocampus and LTC (see also **Reply 1.4.3**). This suggests that their impact on behavioral performance is less pronounced than that of the EC.

Finally, in the **Discussion** section, we cautiously address the contributions of the hippocampus, EC, and LTC to WM processing, in line with previous reports linking WM processing to elevated activity in the hippocampus ^{3, 4}, EC ⁵, and LTC ⁶.

Action: We have carefully addressed the contributions of the hippocampus, EC, and LTC to WM processing. The revised text now reads:

The current results show that decoding accuracy on WM load, derived from power features in the hippocampus, EC, or LTC, was significantly above chance, indicating that these regions contribute to WM processing. These findings are consistent with previous reports linking WM processing to elevated activity in the hippocampus ^{3, 4}, EC ⁵, and LTC ⁶. ...

1.4.3 Comparison of power features between high- and low-performing individuals in hippocampus and lateral temporal cortex

An important aspect of the current research was the comparison of the power features between high- and low-performing individuals. Why were these analyses only conducted on entorhinal cortex, and not on hippocampus or lateral temporal cortex? Would those areas show similar associations with behavioral performance? As decoding is a very powerful approach, this type of cross-validation would be helpful to understand the specificity of the reported effects.

Reply: We agree that it is important to verify whether the association between power features and behavioral performance is unique to the EC. Accordingly, we conducted additional analyses to compare the decoding results of the hippocampus and LTC between high- and low-performing individuals. In contrast to the EC, decoding accuracy and cross-regional generalization in these regions did not differ significantly between performance groups. These findings further underscore the specific functional role of EC power features in WM performance.

Action: We have now added analyses in the **Results** section comparing high- and low-performing individuals in both the hippocampus and LTC. It now reads:

EC Activity Pattern Linked to Better Memory Performance

The single-regional and cross-regional decoding accuracies were significantly above chance level in both the high- and low-performing groups for the EC (low-group: single-regional $53.21 \pm 3.24\%$, cross-regional $52.57 \pm 2.17\%$; high-group: single-regional $56.22 \pm 2.87\%$, cross-regional $54.39 \pm 2.33\%$), hippocampus (low-group: single-regional $53.89 \pm 3.16\%$, cross-regional $51.53 \pm 2.81\%$; high-group: single-regional $54.12 \pm 3.32\%$, cross-regional $51.95 \pm 2.46\%$), and LTC (low-group: single-regional $52.18 \pm 3.47\%$, cross-regional $51.64 \pm 2.72\%$; high-group: single-regional $51.76 \pm 4.60\%$, cross-regional $51.36 \pm 2.13\%$; **Supplementary Table S2**). The further comparison revealed that single-regional (**Fig. 5B**) and cross-regional (**Fig. 5C**) decoding accuracies were significantly higher for the high-performing group compared to the low-performing group in the EC (single-regional: $t = 7.15$, $p < 0.001$; cross-regional: $t = 5.94$, $p < 0.001$), whereas decoding accuracies did not differ significantly between performance groups in the hippocampus (single-regional: $t = 0.49$, $p = 0.621$; cross-regional: $t = 1.14$, $p = 0.264$) or LTC (single-regional: $t = -0.74$, $p = 0.457$; cross-regional: $t = -0.84$, $p = 0.406$). These findings further underscore the specific functional role of the EC in WM performance.

Fig. 5 (B) Comparison of single-regional decoding accuracy between high- (dark color) and low-performing groups (light color) using features from the EC (blue), hippocampus (red), and LTC (green) to decode load 6 vs load 8. *** $p < 0.001$. (C) Comparison of cross-regional decoding accuracy between high- (dark color) and low-performing groups (light color) using features from the EC (blue), hippocampus (red), and LTC (green). *** $p < 0.001$.

1.4.4 Phase-locking connectivity analysis between hippocampus and lateral temporal cortex

Similarly, why were the phase-locking value connectivity analyses only conducted between the entorhinal cortex and other areas, and not between hippocampus and lateral temporal cortex? It would be interesting to see whether the load-dependent connectivity increase is specific to the entorhinal cortex or something generally seen between all inspected areas.

Reply: We agree that it is important to determine whether the load-dependent increase in PLV connectivity is specific to the EC. We first examined the effect of WM load on hippocampus-LTC connectivity and observed a significant main effect (repeated-measures analysis of variance, $F(2,24) = 7.26$, $p = 0.02$). However, when we controlled for the influence of EC's connectivity, the PLV between the hippocampus and LTC was no longer associated with WM load (all $ps > 0.05$). These results suggest that the observed load-dependent increase in connectivity is specific to the EC.

Action: We have now described the hippocampus-LTC PLV and EC-residualized PLV in the **Results** section:

To assess whether the load-dependent connectivity increase is specific to the EC, we first examined the effect of WM load on hippocampus-LTC connectivity and observed a significant main effect ($F(2,24) = 7.26, p = 0.02$; post-hoc tests: load 4 vs load 6, $p = 0.12$; load 4 vs load 8, $p = 0.01$; load 6 vs load 8, $p > 0.05$). Next, we removed the EC's influence by using EC-residualized PLV as the dependent variable; under this analysis, hippocampus-LTC connectivity was no longer significantly associated with WM load (**Fig. 5F**; all $ps > 0.05$). These results indicate that the load-dependent increase in connectivity is specific to the EC.

Fig. 5 (F) The average EC-residualized PLV in the 1-40 Hz range for load 4 (purple), load 6 (yellow), and load 8 (orange).

1.4.5 Sentence modifications

In line 317 the sentence starts And... Please correct.

In line 323 the abbreviation SVM is defined for the second time.

Lines 337-338: "Across all participants, The median memory capacity across all participants..." Please correct.

Line 540: "Result, Results showed significant..." Please correct.

Action: We thank the Reviewer for identifying these typos, which helped us improve the clarity and precision of our manuscript. We have revised the sentence in line 317 to remove the incomplete start "And," eliminated the redundant definition of "SVM" in line 323, and removed repeated words in lines 337-338 and line 540. We have also carefully proofread the entire manuscript to prevent similar errors in the future.

1.4.6 Data distribution

Figure 2A left. The distributions of the data do not look very nice. Could the authors comment on that.

Reply: We thank the Reviewer for raising this point of data distribution. We guess the Reviewer is referring to **Fig. 2D** (left) (shown below), where several lower decoding accuracy values are observed. We suspect these lower values result from the random trial selection inherent in our cross-validation procedure, where each of 100 iterations randomly selects 70% of trials for training and 30% for testing, thus occasionally leading to suboptimal model training. To ensure that these lower values were not due to the limited number of iterations, we repeated the analysis with 150 iterations. The resulting distribution was similar, with only a few lower accuracy instances (see **Fig. R1**), confirming that our results are robust and not unduly affected by the number of iterations or isolated extreme values.

Fig. 2 (D) Single-regional decoding accuracy on WM load

Fig. R1 Data distribution with 100 and 150 iterations

1.4.7 Summaries and interpretation in results section

Overall, in the results section, quite a bit of text is devoted to summaries of the key findings that already contain a bit of interpretation (end of each result section).

Maybe extensive summaries with interpretation are not needed if the results are described in a clear way from the beginning.

Action: Following this valuable suggestion, we have removed the extensive interpretation in the **Results** section to avoid redundancy.

1.5 Discussion

1.5.1 Clarification of Description

The sentence in lines 551-553 "Cross-regional and residual-based decoding analyses indicated that the EC shares more WM relevant information with the other two regions under medium-to-high load" could easily be interpreted so that EC shares more information under medium-to-high load than under low-to-medium load, opposed to that EC shares more information compared with the other regions as it probably should say based on the results. Some clarification could be helpful here.

Reply: We thank the reviewer for the suggestion to clarify this point. We have now more clearly illustrated this result.

Action: We have revised this sentence:

... Cross-regional and residual-based decoding analyses indicated that the EC shares more WM-relevant information **compared** with the other two regions under medium-to-high load condition. ...

1.5.2 The role of hippocampus and LTC in WM maintenance

In lines 558-559 it is said that "A key issue to address is why the EC, but not the hippocampus or LTC, contributes to the active maintenance of medium-to-high load?" Consistent with our previous point (see the comments for the results section), it was not clear whether the role of the hippocampus or LTC in WM maintenance was actually statistically tested. Is it then correct to say that they do not contribute to WM maintenance in medium-to-high loads?

Reply: We totally agree that our findings indicate that under medium-to-high load conditions, the EC exhibits a significantly greater contribution than the hippocampus and LTC, rather than playing an exclusive role.

Action: We have clarified this point accordingly in the **Discussion** section. It now reads:

A key issue to address is why the EC contributes more to WM maintenance than the hippocampus or LTC under medium-to-high load condition.

1.5.3 Discussion on previous WM load studies and connectivity findings

Some previous WM load studies are presented in the third paragraph of the discussion (line 576 onwards). From there, one could be left with the impression that not many studies exist where WM load effects were studied beyond local activation patterns. This can feel a bit misleading, considering also that the authors have cited several system-level and/or connectivity studies about WM load effects (including e.g. Wang et al. 2024; Zhao et al. 2024; Van Snellenberg et al. 2015 - see also our previous comment about the introduction) and that previous studies, presumably from the same dataset as the current study, have shown WM load effects in the connectivity between the hippocampus and entorhinal cortex (Li et al. 2024) and between the hippocampus and cortex (Boran et al. 2019). Both studies are cited in the manuscript. The reader could benefit from a more elaborated discussion of these previous studies, including the studies conducted previously from this same dataset.

Reply: We thank the Reviewer for encouraging us to more clearly express our contributions relative to existing literature. While previous studies have examined the activation and connectivity of multiple brain regions in relation to WM load, our study extends this work in several important ways. In addition to investigating each brain region individually, we compared their contributions to WM processing, examined how these contributions change across different load levels, and integrated inter-regional connectivity into our analysis. This comprehensive approach offers a connectivity-based interpretation of the observed effects and provides an integrated perspective on how brain regions coordinate to support WM processing under varying load conditions.

Action: Following the Reviewer' suggestion, we have now thoroughly re-written this paragraph. It now reads:

Previous studies have examined the relationship between brain activation, connectivity, and varying WM loads using single-unit recordings, intracranial and scalp EEG, and fMRI. Specifically, single-unit recordings have shown that

hippocampal firing during WM maintenance increases from low-to-medium load but does not continue to rise from medium-to-high load ³. fMRI studies have observed a progressive increase in executive control network activation ⁷ and an inverted-U-shaped activation pattern ⁸ across multiple brain regions with higher WM load. Additionally, electrophysiological studies have reported enhanced fronto-parietal connectivity ⁹ and stronger theta/alpha band hippocampus-EC connectivity ¹⁰ as WM load increases. Our study introduces an alternative approach by integrating between-region comparisons and inter-regional connectivity within a unified framework. The results suggest that the brain reallocates neural resources to meet varying cognitive demands. The EC, acting as a primary gateway between the hippocampus and neocortex and exhibiting load-dependent increases in connectivity, contributes more than the other two regions under higher cognitive demands.

1.5.4 Language review

Finally, the language could be improved, and typos carefully checked throughout the manuscript.

Action: We have carefully reviewed the manuscript to correct any typos and have made necessary revisions to improve the overall clarity and precision of the language.

1.6 Reporting summary

1.6.1 Trial exclusion and visual inspection

This report states shortly that no data were excluded, but the manuscript describes trial exclusion that was based on the visual inspection. Please check the consistency, expand how the visual inspection was done, and include information of the final number of included (correct) trials to the manuscript.

Action: We have revised the **Reporting summary** to ensure consistency regarding trial exclusion:

We excluded 140/3057 trials due to artifacts, typically caused by large singular artifacts from cable movement or signal drift.

Furthermore, we have provided a detailed description of the visual inspection procedure in the **Data acquisition and preprocessing**, along with the final numbers of included correct trials:

... We systematically inspected the raw data to identify and exclude trials with residual artifacts, including signal drift or large singular artifacts typically caused by cable movement. As a result, 65 trials were excluded for load 4 (5.5%), 36 trials for load 6 (3.9%), and 39 trials for load 8 (5.1%) across all participants. Subsequent analyses were performed on the correct trials (1054 trials for load 4, 777 trials for load 6, and 747 trials for load 8). ...

2 Reviewer #2:

In this manuscript, the authors seek to test how the brain flexibly adapts to different levels of working memory (WM) load. They conducted a Sternberg working memory task in patients with intracranial electrodes and specifically investigated modulation of spectral signals in entorhinal cortex (EC), hippocampus, and lateral temporal cortex (LTC) in response to low (4 letters), medium (6), and high (8) WM loads. The authors report a series of complimentary analyses demonstrating a specific contribution of EC to WM maintenance under the high (relative to medium) load condition. Signal from all three regions can be used to decode low vs. medium and medium vs. high load conditions; however, when overlapping data from EC is regressed out of the hippocampal and LTC data, decoding accuracy is significantly reduced in those regions. Cross-region decoding indicates that there is shared information between EC and hippocampus and between EC and LTC. EC decoding accuracy is higher in participants with high compared to low performance. Phase locking values between EC and hippocampus and between EC and LTC are significantly modulated by WM load. The authors conclude that the EC may serve WM in the same manner as it does in LTM, namely by processing general information prior to sending this information to the hippocampus and relaying hippocampal outputs to the cortex for storage and retrieval.

Overall, this is a well written paper with robust and compelling findings. Each analysis provides important and complimentary insights into the role of EC in WM maintenance which will be of interest to the field. I have a few comments and points of clarification which I outline below.

Reply: We thank the reviewer for this appreciation.

2.1 Trial counts for classification

As expected, the number of trials differs across load condition (Lines 210-211, "load 4: 1054 samples, load 6: 777 samples, load 8: 747 samples"); however, it is not clear from the methods whether measures were taken to address this imbalance in the classification procedure. The methods mention that 70% of trials were used as training data, but the trial counts may still be imbalanced which can bias classification performance.

Reply: We thank the reviewer for highlighting this important concern. We agree that the potential impact of trial count imbalance on classification performance must be addressed. To do so, we randomly selected trials from load 4 and load 6 to match the number of trials in load 8. Then, we repeated all decoding analyses, including single-regional decoding, cross-regional generalization and residual-based decoding, using these balanced datasets. This resampling was performed 10 times, with statistical comparisons between brain regions conducted for each iteration. Our results were consistently replicated across all iterations: the EC showed both higher decoding accuracy under medium-to-high load and superior cross-regional generalization, and removing EC-related information significantly reduced residual decoding accuracy in the hippocampus and LTC (**Fig. S1**). Thus, we believe that our findings are not driven by imbalanced trial counts.

Action: We have now added decoding results based on balanced trial counts in the **Results** section:

Decoding Results Based on Balanced Trial Count

To rule out the potential impact of trial count on classification performance, we randomly selected 747 trials from load 4 and load 6 to match the number of trials in load 8. This resampling procedure was repeated 10 times, and all decoding analyses were re-executed with balanced trial counts. For each resampling, statistical comparisons between brain regions were conducted separately. Our results revealed no significant differences in decoding accuracy (**Supplementary Fig. S1A**) or cross-region generalization (**Supplementary Fig. S1C**) among brain regions under low-to-medium load condition. Under medium-to-high load condition, the EC demonstrated the highest decoding accuracy (**Supplementary Fig. S1B**) and cross-region generalization (**Supplementary Fig. S1D**). Moreover, removing EC-shared information significantly reduced the decoding accuracy of the hippocampus and LTC (**Supplementary Fig. S1E**). In summary, these results are consistent with the original findings, demonstrating that the EC exhibits enhanced decoding performance and shares load-related information with other regions under medium-to-high load condition.

Supplementary Fig. S1 Decoding results with balanced trial counts. (A) Left: Mean single-regional decoding accuracy on low-to-medium load across 10 resampling iterations with balanced trial counts. Right: Distribution of t-values for pairwise decoding accuracy comparisons across brain regions. (B) Left: Mean single-regional decoding accuracy on medium-to-high load across 10 resampling iterations with balanced trial counts. Right: Distribution of t-values for pairwise decoding accuracy comparisons across brain regions. $***p < 0.001$. (C) Left: Mean cross-regional decoding accuracy on low-to-medium load across 10 resampling iterations with balanced trial counts. Right: Distribution of t-values for pairwise decoding accuracy comparisons across brain regions. (D) Left: Mean cross-regional decoding accuracy on medium-to-high load across 10 resampling iterations with balanced trial counts. $***p < 0.001$. Right: Distribution of t-values for pairwise

decoding accuracy comparisons across brain regions. (E) Left: Comparison between the original decoding accuracy (dark color) and EC-residual decoding accuracy (light color) over 10 resampling iterations. *** $p < 0.001$. Right: Distribution of t-values for comparisons between original (orig) and EC-residual (residual) decoding accuracy over 10 resampling iterations. The source data are provided in the Source data file.

2.2 Features used for cross-regional decoding analysis

It is not always clear which features are used for classification. Are the PCA features only used for the within-region classification analysis? The cross-region analysis references "power features" which implies that non-PCA features were used which seems potentially in danger of over fitting given the very large number of features.

Reply: We apologize for not clearly conveying this in our previous version; we used PCA features in the cross-regional decoding analysis.

Action: We have revised the sentence to clearly state that PCA features were used in the cross-regional decoding analysis. It now reads:

... Taking the EC as an example, we first applied PCA to the EC power data and retained principal components that explained at least 99% of the variance. The transformed EC data served as the training set to decode load 4 vs load 6. Next, we applied the same PCA transformation matrix, previously fitted to the EC training data, to the hippocampus and LTC power data. The transformed hippocampus and LTC data were then separately used as test sets to evaluate the model. ...

2.3 Description of the WM network

The authors state that, "Our findings support the notion that increased WM demands require a more integrated, less modular network configuration" (lines 582-583) and reference "the WM network" (line 607). These statements seem to be a bit of an over-interpretation of the data given that the authors did not directly measure modularity and the regions considered here do not encompass the entirety of the regions recruited during WM maintenance (and thus cannot be considered "the WM network").

Reply: We thank the reviewer for the suggestion that made our wording more rigorous.

Action: We have removed the statements regarding modularity and network and revised the wording more cautiously. Specifically, We have revised "... more integrated, less modular network configuration" to: ... Our study introduces an alternative approach by integrating between-region comparisons and inter-regional connectivity within a unified framework. ...

We have revised "...the WM network..." to:

In summary, the brain is not static but dynamically adapts to cognitive load and behavioral demands, with the EC playing a crucial role in this process.

2.4 Significance of classifier performance

The authors show that EC generalization outperforms hippocampal generalization (Lines 400-401), but is the EC generalization performance itself significantly above chance? Relatedly, do the LTC-residualized hippocampus and hippocampal-residualized LTC classifiers perform above chance?

Reply: We thank the reviewer for the suggestion to confirm that each region's performance is significantly above chance before making comparisons between regions. Therefore, we assessed the significance of each region's performance separately, and our analyses confirmed that all decoding results were significantly above chance in both the cross-regional and residual-based decoding analyses.

Action: Accordingly, we have added detailed descriptions of the statistical significance tests for each region in both the cross-regional and residual-based decoding analyses, along with the corresponding results, to the **Results** section.

In Greater Load Sensitivity of EC under Medium-to-High Demands:

... To assess the statistical significance of the decoding results, we created a null distribution of the decoding accuracy by shuffling the relationship between labels and data 100 times. Decoding accuracy exceeding the 95% threshold of this null distribution was considered statistically significant. The results showed that the decoding accuracies on low-to-medium and medium-to-high load conditions using power features from the hippocampus, EC, and LTC were significantly above chance level. The detailed 95% threshold values are provided in **Supplementary Table S2**. ...

and in **Higher Cross-Regional Generalization of EC under Medium-to-High Demands:**

The cross-regional decoding accuracies on low-to-medium and medium-to-high load conditions across hippocampus, EC, and LTC were significantly above chance level (see **Supplementary Table S2** for details). ...

and in **Significant Decoding Accuracy Reduction after Removing EC Information:**

... the residual-based decoding accuracies were significantly higher than chance level for the hippocampus ($50.34 \pm 1.33\%$) and the LTC ($50.31 \pm 1.41\%$; see **Supplementary Table S2** for details). ...

... The analysis showed that decoding accuracies remained significantly above chance level for LTC-residualized ($51.36 \pm 1.22\%$) in the hippocampus and the Hipp-residualized ($50.78 \pm 1.48\%$; **Supplementary Table S2**) in the LTC. ...

and in **EC Activity Pattern Linked to Better Memory Performance**

The single-regional and cross-regional decoding accuracies were significantly above chance level in both the high- and low-performing groups for the EC (low-group: single-regional $53.21 \pm 3.24\%$, cross-regional $52.57 \pm 2.17\%$; high-group: single-regional $56.22 \pm 2.87\%$, cross-regional $54.39 \pm 2.33\%$), hippocampus (low-group: single-regional $53.89 \pm 3.16\%$, cross-regional $51.53 \pm 2.81\%$; high-group: single-regional $54.12 \pm 3.32\%$, cross-regional $51.95 \pm 2.46\%$), and LTC (low-group: single-regional $52.18 \pm 3.47\%$, cross-regional $51.64 \pm 2.72\%$; high-group: single-regional $51.76 \pm 4.60\%$, cross-regional $51.36 \pm 2.13\%$; **Supplementary Table S2**). ...

and in **Supplementary Table S2:**

Supplementary Table S2. Chance level for all decoding analyses.

		EC	Hippocampus	LTC
Single-regional (%)	load 4 vs load 6	53.16	52.37	50.37
	load 6 vs load 8	51.05	50.73	51.02

Cross-regional (%)	load 4 vs load 6	53.27	50.42	57.25
	load 6 vs load 8	50.07	50.21	50.63
Residuals-based (%)	EC-residual	/	50.02	50.11
	Control analysis	/	50.31	50.05
Single-regional groups (%)	low group	50.24	50.98	50.57
	high group	50.13	50.20	50.18
Cross-regional groups (%)	low group	50.29	50.16	50.48
	high group	50.12	50.12	50.14

2.5 PLV between hippocampus and LTC

Does the PLV between hippocampus and LTC vary as a function of load? (It might be informative to perform the phase synchrony analysis on the EC-residualized data.)

Reply: We agree that it is important to determine whether the load-dependent increase in PLV connectivity is specific to the EC. We first examined the effect of WM load on hippocampus-LTC connectivity and observed a significant main effect (repeated-measures analysis of variance, $F(2,24) = 7.26$, $p = 0.02$). However, when we controlled for the influence of EC's connectivity, the PLV between the hippocampus and LTC was no longer associated with WM load (all $ps > 0.05$). These results suggest that the observed load-dependent increase in connectivity is specific to the EC.

Action: We have now described the hippocampus-LTC PLV and EC-residualized PLV in the **Results** section:

To assess whether the load-dependent connectivity increase is specific to the EC, we first examined the effect of WM load on hippocampus-LTC connectivity and observed a significant main effect ($F(2,24) = 7.26$, $p = 0.02$; post-hoc tests: load 4 vs load 6, $p = 0.12$; load 4 vs load 8, $p = 0.01$; load 6 vs load 8, $p > 0.05$). Next, we removed the EC's influence by using EC-residualized PLV as the dependent variable; under this analysis, hippocampus-LTC connectivity was no longer

significantly associated with WM load (**Fig. 5F**; all $ps > 0.05$). These results indicate that the load-dependent increase in connectivity is specific to the EC.

Fig. 5 (F) The average EC-residual PLV in the 1-40 Hz range for load 4 (purple), load 6 (yellow), and load 8 (orange).

2.6 EC electrode placement

Can the authors clarify this seeming discrepancy: on lines 157-159, the authors state that "Targeted regions ... included .. the EC in the left ($n = 12$) and right ($n = 11$) hemisphere" which seems to imply that one of the 13 participants did not have any EC electrodes. However, on lines 165-166, the authors state that there were " 3.5 ± 0.8 channels per participant in the EC (with a range of 2 to 4)."

Reply: Thank you for pointing out this discrepancy. We apologize for the confusion. As illustrated in **Fig. R2**, all 13 participants had electrodes implanted in the EC. Specifically, ten participants had electrodes in both hemispheres, two had electrodes only in the left EC, and one had electrodes only in the right EC.

Action: We have re-written this sentence to clearly describe the hemispheric distribution of the EC electrodes:

... Ten participants had electrodes in the EC of both hemispheres, while two participants had one electrode implanted in the left EC, and one participant had one electrode in the right EC. ...

Fig. R2 EC Electrode Placement across Hemispheres

2.7 The number of sessions and principal components

In general, the level of detail around numbers of trials, electrodes, etc. is excellent and the authors provide means, minimums, and maximums. The same should be provided for the number of sessions (p 6) and the number of principal components (line 222) as currently the authors state, "with some participants undertaking multiple sessions, up to a maximum of eight" and that there were "several principal components."

Action: Following the Reviewer' comment, we have now provided detailed information on the number of sessions and principal components in the **Methods** Section.

We have described the number of sessions in the **Task Design and Behavioral Metrics:**

... Each participant conducted multiple sessions across several recording days, with an average of 4.8 ± 1.9 sessions (ranging from 2 to 8).

We have described the number of principal components in the **Single-regional decoding:**

... Meanwhile, to reduce the feature dimensionality, principal component analysis (PCA) was applied to the training data set to keep several principal components (K components; 5.1 ± 6.8 components, ranging from 1 to 25) that explained at least 99% of the variance in the data. ...

2.8 Description of Figure 1B reference

I found lines 101-103 "Our findings show..." a little confusing because this statement references figure 1B which does not show data but a schematic/predicted results.

Action: We have revised the statement referencing Figure 1B, which presents a schematic framework for decoding analysis using single-regional power features. It now reads:

... Decoding analyses were conducted using single-regional power features (**Fig. 1B**), cross-regional decoding generalization (**Fig. 1C**), and residual-based decoding to remove EC-shared activity (**Fig. 1D**). ...

2.9 Description of task

The authors state on lines 129-130, "We adopted a modified Sternberg task that temporally dissociated the processes of WM encoding, maintenance, and retrieval." I found the 'temporal dissociation...' phrase confusing, don't all Sternberg (and all WM tasks) temporally dissociate study/maintenance/test?

Action: We have removed the phrase and simply stated: We adopted a modified Sternberg task (**Fig. 2A**).

2.10 Parameter details for phase locking value

In a few cases some of the terminology was a bit underspecified. For instance, the authors state that they used "consistent parameters for time-frequency analysis" (lines 275-276), but it's unclear what "consistent" means in this context.

Action: We have clarified the PLV analysis parameters in the **Methods** section. It now reads:

... The signal was convolved with complex Morlet wavelets (six cycles) across frequency from 1 to 40 Hz (in 1 Hz steps) with a time resolution of 1 ms. ...

3 Reviewer #3:

This is a very interesting manuscript evaluating entorhinal cortex, hippocampal, and lateral temporal cortex activity during varying working memory tasks in patients with drug-resistant epilepsy. The manuscript is well communicated with valid, relevant methods and robust interpretation of results. While this study has great strengths, the authors don't mention the limitations of studying memory function in patients with drug-resistant epilepsy. In addition, there is information that is missing, mostly in the Methods section, which would greatly improve the applicability and generalizability of the manuscript that are listed below.

Reply: We thank the reviewer for this appreciation.

3.1 Clinical features

There is no information regarding the patient's clinical features, including the following:

- 1.Side of recording (did all patients have bilateral recordings?)
- 2.MRI findings (i.e. hippocampal sclerosis)
- 3.Pathology findings
- 4.Medications for each patient at the time of testing

Reply: We appreciated this valuable suggestion. In the current study, all patients had bilateral recordings and had their normal anti-seizure medication at the time of testing. Additionally, the pathology findings for each patient have been described in a table.

Action: We have now included the patient's clinical features.

Regarding bilateral recordings, we now report in **Electrode contact localization and selection:**

... Specifically, all thirteen participants had electrodes implanted in the hippocampus of both the left and right hemispheres. Ten participants had electrodes in the EC of both hemispheres, while two participants had one electrode implanted in the left EC, and one participant had one electrode in the right EC. Across all our participants, LTC contacts were selected from the same electrode as the hippocampus and EC contacts. As a result, all thirteen participants had LTC recording sites in both the left and right hemispheres. ...

Information on pathology, medication, and other relevant details is included in the **Participants** and **Supplementary Table S1**:

Participants. ...We listed the patients' pathology and seizure onset zone (SOZ) in **Supplementary Table S1**. All patients had their normal anti-seizure medication at the time of testing. ...

Supplementary Table S1. Subject characteristics.

Patient	Age [y]	Sex	Pathology	Seizure onset zone	
1	24	f	Xanthoastrozytoma WHO II	Hippocampus	right
2	39	m	Hippocampal gliosis	Hippocampus	right
3	18	f	Hippocampal sclerosis	Hippocampus	left
4	28	m	Posttraumatical lesion	Hippocampus	bilateral
5	31	m	Hippocampal sclerosis	Amygdala	right
6	47	m	Hippocampal sclerosis	Hippocampus	right
7	56	f	mesial temporal sclerosis	Entorhinal cortex	right
8	19	f	mesial temporal sclerosis	Amygdala	right
9	35	m	non-lesional	Hippocampus	bilateral
10	51	f	Hippocampal sclerosis	Hippocampus	left
11	30	m	Dysembryoplastic neuroepithelial tumour	Hippocampus	left
12	29	f	non-lesional	Hippocampus	bilateral
13	56	m	Hippocampal sclerosis	Hippocampus	bilateral

3.2 Lateralizing features on WM

Since memory has lateralizing features (i.e. the left hemisphere is associated with episodic verbal memory), the authors should compare how the side of the recording affects WM performance or at least add side as a covariate.

Reply: We fully agree that hemispheric differences should be considered, especially given the left hemisphere's association with episodic verbal memory. To address this, we conducted two analyses. First, we divided the electrodes into left and right hemisphere groups and re-ran the decoding analysis for each group. Our original results were replicated in both groups: for low-to-medium load, the EC showed comparable decoding accuracy to the hippocampus and LTC, while for medium-to-high load, the EC exhibited higher decoding accuracy. Second, since our decoding results are based on power, we compared power across hemispheres and also examined inter-regional PLVs between hemispheres. No significant hemispheric

effects were observed, and we discuss the potential impact of hemispheric differences on memory in the **Discussion** section.

Action: We have included these analysis results in the **Results** section:

Hemispheric Effects on Decoding, Power, and Connectivity

To examine the impact of hemisphere on our findings, we performed additional analyses on the participants with bilateral electrode implantation in the hippocampus, EC, and LTC (n = 10). We conducted single-regional decoding analysis using the power features from the left and right hemispheres separately. The results revealed no significant differences in decoding accuracy on low-to-medium load among the three brain regions in either the left hemisphere (hippocampus: 59.06%, EC: 59.22%, LTC: 59.29%; all p s > 0.05) or the right hemisphere (hippocampus: 59.32%, EC: 59.39%, LTC: 59.40%; all p s > 0.05). Under medium-to-high load condition, decoding accuracy using EC power features was significantly higher in both the left hemisphere (hippocampus: 51.92%, EC: 54.93%, LTC: 52.46%; all p s < 0.001) and the right hemisphere (hippocampus: 50.72%, EC: 53.09%, LTC: 51.47%; all p s < 0.001). We thus did not find the hemispheric lateralization in the adaptation to WM load in these brain regions.

As a further test, we compared the power between the left and right hemispheres for each brain region, since our decoding analyses are based on power. The analysis revealed no significant hemispheric differences in power across the hippocampus ($p = 0.33$), EC ($p = 0.06$), or LTC ($p = 0.35$). Finally, we examined inter-regional connectivity between the left and right hemispheres and found no significant differences in the connectivity of EC-hippocampus ($p = 0.94$), EC-LTC ($p = 0.21$), or hippocampus-LTC ($p = 0.29$). In summary, the current study did not find the influence of hemisphere.

We have also discussed the potential impact of the hemisphere effect on the results in the **Discussion** section:

The left hemisphere of the brain is generally considered to play a dominant role in verbal memory processing¹¹. As one would expect for this language related task, a previous analysis of scalp EEG data for the same task found a clear lateralization to the left cortical hemisphere¹². However, there was no left-hemispheric predominance in the MTL^{3,12}. Since we record from patients with MTL epilepsy, we

cannot rule out plastic reorganization resulting in bilateral MTL processing of this task.

3.3 Epileptogenicity

Information should be given on the epileptogenicity of all recording sites (i.e. frequency of epileptogenic discharges and seizure activity/seizure onset zone -if identified).

Reply: We thank the reviewer for this important suggestion.

Action: We have now included information on the seizure onset zone in the **Supplementary Table S1**, as noted in our response to **Comment 3.1** above.

3.4 Limitations and generalizability

The authors should discuss the shortcomings of studying electrical activity in this patient population and its applicability in the general population.

Reply: We thank the reviewer for the valuable suggestion. Accordingly, we have now expanded the **Discussion** section to address potential limitations of studying an epilepsy patient population.

It should be noted that our data were recorded in patients with epilepsy, which could affect our results. To mitigate this confounding factor, we carefully screened all trials and excluded those with signs of epileptiform activity before analysis. Still, the lack of left-hemispheric dominance might result from plastic reorganization of the MTL. However, several of our other observations seem to be independent of epilepsy. First, the low-frequency power dynamics in these regions reflect cognitive processes: they increase during encoding, remain stable during maintenance, and decline during retrieval, consistent with previous reports^{3, 12, 13} and are unlikely to result from epileptiform patterns¹⁴. Second, both power and PLV are modulated by WM load: power from all three regions successfully decoded WM load, and connectivity exhibited a load-dependent increase, effects that speak for association with cognitive brain function. Third, comparisons of power and PLV between electrodes within and outside the SOZ revealed no significant differences, indicating that this brain tissue was associated with brain function even though the tissue was subsequently resected. Together, these functional associations suggest that our findings might be generalizable to the general population.

4 Reviewer #4:

Reply: We thank the reviewer for the diligent effort and valuable feedback, which have significantly improved the quality of our manuscript.

Reference

1. Fascianelli V, Battista A, Stefanini F, Tsujimoto S, Genovesio A, Fusi S. Neural representational geometries reflect behavioral differences in monkeys and recurrent neural networks. *Nature communications* **15**, 6479 (2024).
2. Li J, *et al.* Functional specialization and interaction in the amygdala-hippocampus circuit during working memory processing. *Nature communications* **14**, 2921 (2023).
3. Boran E, *et al.* Persistent hippocampal neural firing and hippocampal-cortical coupling predict verbal working memory load. *Science advances* **5**, eaav3687 (2019).
4. Rutishauser U, Reddy L, Mormann F, Sarnthein J. The Architecture of Human Memory: Insights from Human Single-Neuron Recordings. *The Journal of neuroscience : the official journal of the Society for Neuroscience* **41**, 883-890 (2021).
5. Newmark RE, Schon K, Ross RS, Stern CE. Contributions of the hippocampal subfields and entorhinal cortex to disambiguation during working memory. *Hippocampus* **23**, 467-475 (2013).
6. Singh B, *et al.* Brain-wide human oscillatory local field potential activity during visual working memory. *iScience* **27**, 109130 (2024).
7. Zhao W, *et al.* Activity flow under the manipulation of cognitive load and training. *NeuroImage* **297**, 120761 (2024).
8. Thomas ML, Duffy JR, Swerdlow N, Light GA, Brown GG. Detecting the Inverted-U in fMRI Studies of Schizophrenia: A Comparison of Three Analysis Methods. *Journal of the International Neuropsychological Society : JINS* **28**, 258-269 (2022).
9. Wang Y, *et al.* Fronto-parietal activity changes associated with changes in working memory load: Evidence from simultaneous electroencephalography and functional near-infrared spectroscopy analysis. *The European journal of neuroscience* **60**, 5413-5427 (2024).
10. Li J, Cao D, Li W, Sarnthein J, Jiang T. Re-evaluating human MTL in working memory: insights from intracranial recordings. *Trends in cognitive sciences*, (2024).
11. Kubota E, Grill-Spector K, Nordt M. Rethinking cortical recycling in ventral temporal cortex. *Trends in cognitive sciences* **28**, 8-17 (2024).
12. Dimakopoulos V, Mégevand P, Stieglitz LH, Imbach L, Sarnthein J. Information flows from hippocampus to auditory cortex during replay of verbal working memory items. *eLife* **11**, (2022).
13. Li J, *et al.* Anterior-Posterior Hippocampal Dynamics Support Working Memory Processing. *The Journal of neuroscience : the official journal of the Society for Neuroscience* **42**, 443-453 (2022).
14. Boran E, Stieglitz L, Sarnthein J. Epileptic High-Frequency Oscillations in Intracranial EEG Are Not Confounded by Cognitive Tasks. *Frontiers in human neuroscience* **15**, 613125 (2021).

Reviewer comments

We thank all expert Reviewers for the thorough review and positive feedback on our revised manuscript. Each reviewer made suggestions that would greatly strengthen the manuscript. The responses to Reviewer's comments are shown in blue font.

1 Reviewer #1:

Authors have carefully addressed all my concerns.

Reply: We thank the reviewer for their time and effort to review our manuscript, and for their positive feedback.

2 Reviewer #2 (Remarks to the Author):

The authors have addressed the majority of my concerns. My only remaining point is that the balanced classification analysis currently described in the supplementary should be moved to the main text and replace the imbalanced classification analysis, as it is not valid to conduct classification when classes are imbalanced.

Reply: Using the full dataset with all trials maximizes statistical power and avoids selection bias. As suggested by the reviewer, the balanced control analysis with reduced trial numbers certainly added value to our manuscript. However, as not to confuse readers, we prefer to present this control analysis rather in the supplementary material than in the main text.

3 Reviewer #3

3.1 Remarks to the Author:

The authors fully addressed concerns in the revision.

Reply: We thank the reviewer for their time and effort to review our manuscript, and for their positive feedback.

3.2 Remarks on code availability:

I have reviewed the code in the link above, and it appears to be suitable for its intended purpose. I have not installed and run the code.

Reply: We thank the reviewer for their time and effort in evaluating the code.

4 Reviewer #4 (Remarks to the Author):

Reply: We thank the reviewer for their time and effort to review our manuscript, and for their positive feedback.